# A Deployment Method for Motor Fault Diagnosis Application Based on Edge Intelligence

**DOI:** 10.3390/s25010009

**Published:** 2024-12-24

**Authors:** Zheng Zhou, Yusong Qiao, Xusheng Lin, Purui Li, Nan Wu, Dong Yu

**Affiliations:** 1Shenyang Institute of Computing Technology, Chinese Academy of Sciences, Shenyang 110168, China; zhouzheng18@mails.ucas.ac.cn (Z.Z.); qiaoyusong21@mails.ucas.ac.cn (Y.Q.); linxusheng21@mails.ucas.ac.cn (X.L.); lipurui22@mails.ucas.ac.cn (P.L.); wunan@sict.ac.cn (N.W.); 2University of Chinese Academy of Sciences, Beijing 100049, China; 3Shenyang CASNC Technology Co., Ltd., Shenyang 110168, China

**Keywords:** intelligent CNC systems, edge intelligence, fault diagnosis, model deployment

## Abstract

The rapid advancement of Industry 4.0 and intelligent manufacturing has elevated the demands for fault diagnosis in servo motors. Traditional diagnostic methods, which rely heavily on handcrafted features and expert knowledge, struggle to achieve efficient fault identification in complex industrial environments, particularly when faced with real-time performance and accuracy limitations. This paper proposes a novel fault diagnosis approach integrating multi-scale convolutional neural networks (MSCNNs), long short-term memory networks (LSTM), and attention mechanisms to address these challenges. Furthermore, the proposed method is optimized for deployment on resource-constrained edge devices through knowledge distillation and model quantization. This approach significantly reduces the computational complexity of the model while maintaining high diagnostic accuracy, making it well suited for edge nodes in industrial IoT scenarios. Experimental results demonstrate that the method achieves efficient and accurate servo motor fault diagnosis on edge devices with excellent accuracy and inference speed.

## 1. Introduction

With the rapid advancement of Industry 4.0 and intelligent manufacturing, servo motors, core components of the new generation of CNC systems, are playing an increasingly pivotal role [1,2]. The performance of servo motors directly affects the machining precision and production efficiency of CNC systems, making their intelligent diagnosis and maintenance essential to ensuring the efficient operation of manufacturing systems [3]. However, existing servo motor fault diagnosis technologies heavily rely on manual feature extraction and expert experience, which often need help to meet the real-time and high-precision demands of complex and dynamic industrial environments. Particularly in intelligent manufacturing settings, achieving a real-time diagnosis of servo motors under the constrained resources of edge computing remains a critical challenge [4].

Currently, research on motor fault diagnosis primarily focuses on enhancing the accuracy and efficiency of fault detection through traditional deep learning and edge computing technologies. However, these techniques exhibit significant limitations when applied in real-world edge environments. For instance, most existing deep learning models are challenging to deploy efficiently on resource-constrained edge devices due to their high computational complexity, resulting in prolonged inference times that fail to meet real-time requirements [5,6,7]. Moreover, many models struggle to handle dynamically evolving fault patterns and noisy data, particularly in scenarios involving multiple fault types and high-frequency data collection, thereby compromising the robustness and accuracy of diagnosis [8]. To address the constraints of edge devices, researchers have proposed an edge-device collaborative framework that dynamically adjusts task allocation to achieve efficient and robust real-time diagnosis [9]. Consequently, optimizing deep learning models for edge device deployment through techniques like model pruning, quantization, and knowledge distillation is poised to be a key direction for future motor fault diagnosis research [10].

The primary goal of this study is to explore a deployment method for servo motor fault diagnosis applications based on edge intelligence. By integrating deep learning with edge intelligence technologies, we propose a motor fault diagnosis model, particularly leveraging knowledge distillation and model quantization for lightweight deployment in edge environments. This approach aims to facilitate real-time monitoring and efficient fault diagnosis of servo motors by incorporating edge intelligence, thereby addressing the high requirements for real-time performance and diagnostic accuracy in intelligent manufacturing environments. To this end, a model quantization strategy combined with knowledge distillation is adopted to minimize model complexity while preserving the original model’s diagnostic capabilities, ensuring high responsiveness and precise fault detection on edge devices such as the Raspberry Pi 4B.

Through this approach, the study provides a novel pathway for applying servo motor fault diagnosis in intelligent manufacturing, particularly in real-time monitoring and efficient diagnosis in edge environments. This advances servo motor fault diagnosis technology to a new stage of edge intelligence and offers an innovative perspective on edge computing applications in the industrial domain. Empirical research on lightweight models deployed on edge devices like the Raspberry Pi demonstrates that the proposed method can maintain high diagnostic accuracy and rapid response capabilities, even under constrained computational and memory resources, highlighting its practical value in industrial settings. The structure of this paper is designed to systematically introduce the research background, methods, experiments, and results, thereby aiding readers in comprehensively understanding the proposed edge intelligence fault diagnosis approach and its significance in the field of intelligent manufacturing.

The primary contributions of this study are as follows:(1)A motor fault diagnosis system architecture tailored for edge intelligence environments, utilizing edge computing technology to achieve the real-time monitoring of servo motors, thereby enhancing diagnostic response speed and accuracy. This offers a novel motor fault modeling and management methodology in edge environments.(2)A lightweight model approach based on knowledge distillation, enabling the efficient deployment of the MSCNN-LSTM-Attention model on edge devices while balancing high precision with resource efficiency.(3)Integrating post-training quantization techniques to further refine the student model, significantly reducing computational resource consumption and storage requirements and ensuring the efficient inference of the quantized model on edge devices.(4)Empirical research conducted on edge devices that validates the diagnostic accuracy and real-time performance of the lightweight model under resource-constrained conditions, demonstrating the practical value of the approach in industrial field applications.

This research holds significant value in intelligent manufacturing, particularly in providing robust technical support for the real-time monitoring and efficient fault diagnosis of servo motors. This paper’s structure is meticulously designed to systematically introduce and discuss the research content.

The overall structure of this study is divided into six parts, beginning with an introduction that outlines the research background and motivation. The Section 2 reviews the state-of-the-art motor fault diagnosis technologies and the application of edge intelligence. The Section 3 presents the overall architecture and methodology of the motor fault diagnosis system based on edge intelligence. The Section 4 describes the theoretical foundations for model construction, knowledge distillation, and model quantization optimization proposed in this paper. The Section 5 details the system’s experimental design and empirical research results. Finally, the Section 6 summarizes the conclusions and outlines future research directions.

## 2. Related Work

In this section, we review the application of deep learning in motor fault diagnosis, the evolution of edge computing in industrial environments, and its integration with deep learning, with a particular emphasis on using knowledge distillation and model quantization in edge intelligence contexts. We summarize the development trends of these technologies and their current applications in servo motor fault diagnosis, analyzing their limitations in enhancing diagnostic efficiency and addressing the challenges of complex industrial environments. Furthermore, this section highlights the constraints of fault diagnosis in edge environments. It underscores the necessity of further exploration into lightweight, efficient deployment strategies, thereby providing a theoretical foundation and practical guidance for the research direction of this study.

### 2.1. Application of Deep Learning in Fault Diagnosis

In industrial fault diagnosis, deep learning models have been widely employed for extracting and modeling complex signal features, which are pivotal in enhancing diagnostic accuracy and reducing dependence on manual expertise. Various deep learning architectures have demonstrated remarkable performance in this domain, particularly for fault diagnosis in mechanical equipment such as motors. Convolutional neural networks (CNNs), known for their superior feature extraction capabilities, have emerged as one of the critical tools for fault diagnosis. CNNs excel at extracting meaningful features from vibration signals, automatically learning patterns within complex data without requiring manually defined features. For instance, Huang et al. proposed a shallow multi-scale convolutional network integrated with an attention mechanism, achieving an accuracy of 99.86% in bearing fault diagnosis [11]. Additionally, Lu et al. combined CNNs with Transformer models, introducing multi-head attention mechanisms, which significantly enhanced the diagnostic capability of rotating machinery under complex conditions, achieving an average diagnostic accuracy of 97.57% [12]. Long short-term memory (LSTM) networks mainly capture long-term dependencies in time-series data. The application of LSTM in motor fault diagnosis effectively compensates for the shortcomings of traditional methods in utilizing temporal information. For example, Chen et al. proposed a model that integrates CNN, an improved LSTM, and an attention mechanism, effectively balancing time-series features and enhancing diagnostic precision, resulting in a 5.18% improvement in accuracy for fault diagnosis in chemical processes [13]. In applying deep learning, attention mechanisms have been gradually introduced to the fault diagnosis domain to further enhance the model performance. Incorporating attention mechanisms allows models to automatically focus on the most critical parts of fault signals, thereby improving the classification accuracy. For example, Zhang et al. proposed a diagnostic model combining LSTM and attention mechanisms, demonstrating superior performance in detecting multiple fault states, especially when handling complex fault signals [14]. Although the aforementioned deep learning models have achieved significant advancements in fault diagnosis, their deployment in practical applications still faces numerous challenges, particularly under the constrained resources of edge computing environments. Complex deep learning models struggle to operate efficiently when deployed on resource-limited devices. For example, Fu et al. conducted a study using distillation and quantization methods to lighten a CNN model, successfully deploying it on memory-constrained microcontrollers to suit resource-limited industrial environments [15]. These studies provide the theoretical foundations for further exploration in this paper. By integrating techniques such as knowledge distillation and quantization, this study aims to address the challenges of the lightweight deployment of deep learning models on edge devices, thereby adapting them to resource-constrained industrial environments.

### 2.2. Development of Edge Computing in Industrial Applications

The application of edge computing in industrial environments has been continually expanding to enable localized data processing, reduce latency, and enhance data privacy, making it particularly suitable for industrial monitoring and control systems with stringent real-time requirements. In fault diagnosis for critical equipment such as servo motors, edge computing can significantly reduce the data transmission time and network dependency, improving system responsiveness and reliability. For instance, Qian et al. designed a real-time fault diagnosis system based on edge computing, integrating signal acquisition, feature extraction, and identification, completing the entire diagnostic process within 0.25 s, thus achieving rapid motor fault detection [16]. Although this approach enhances the system’s real-time performance, the high complexity of deep learning models still presents challenges for inference speed. To further optimize performance, Ding et al. developed a lightweight multi-scale network that combines edge computing with adaptive pruning techniques, utilizing weight-sharing convolution and depthwise separable convolution modules to achieve high diagnostic accuracy while significantly reducing the computational costs on edge devices [17]. Additionally, Yang et al. proposed an edge-device collaborative diagnostic framework that integrates lightweight neural networks with a dynamic adaptive diagnostic mechanism, compressing model parameters by 384 times while maintaining 100% classification accuracy in the multi-fault diagnosis of servo motors [18]. Zhang et al. developed a distributed edge computing system based on intelligent task reconstruction, employing an optimized real-time workload allocation algorithm that effectively enhances task execution efficiency in industrial edge environments and reduces computational resource wastage [19]. Meanwhile, Huang et al. introduced a lightweight network architecture based on knowledge distillation, transferring diagnostic knowledge from a complex model to a lightweight model, achieving efficient diagnosis in edge computing scenarios while reducing the parameter count by 88 times without compromising diagnostic performance [20]. The application of edge computing technology has dramatically enhanced the real-time performance and processing efficiency of industrial equipment fault diagnosis, particularly in motor fault diagnosis, providing crucial support for efficient and reliable localized monitoring. However, given the limited computational resources of edge devices, further improving the lightweight nature of deep learning models to facilitate edge deployment remains a significant research direction.

### 2.3. Application of Knowledge Distillation and Model Quantization Techniques

To deploy complex deep learning models on edge devices, knowledge distillation and model quantization have become vital strategies for lightweight models. Knowledge distillation effectively reduces the computational complexity of models while maintaining high performance by transferring knowledge from a teacher model to a student model. The concept of knowledge distillation was first introduced by Hinton et al., who used soft labels to guide the student model’s learning, thereby retaining the predictive capabilities of the complex model [21]. This technique has been widely applied in industrial diagnostics to address the challenges posed by resource-constrained edge devices. In the real-time fault diagnosis of servo motors, He et al. proposed a lightweight network approach based on knowledge distillation, employing an improved MobileNet-V3 as the teacher model to transfer knowledge to a simplified student model. This method achieved approximately 170-fold reductions in computational complexity and parameter count while maintaining an accuracy of 94.37% in RV reducer fault diagnosis [22]. This demonstrates that knowledge distillation can significantly reduce hardware resource requirements while preserving diagnostic performance. Furthermore, Liao et al. developed a lightweight network named BearingPGA-Net, which optimizes model computational efficiency through decoupling knowledge distillation and FPGA acceleration. Experimental results show that this method enhances inference speed by 200 times compared to a CPU, with a performance degradation of less than 0.4Model quantization is another crucial technique for lightweight deep-learning models. Zhong et al. proposed a lightweight approach combining knowledge distillation with adversarial training, significantly enhancing the diagnostic efficiency under small-sample conditions by compressing the parameters of the generative adversarial network [23]. Moreover, Xu et al. utilized transfer learning to optimize an online CNN diagnostic framework, significantly improving the efficiency and accuracy of real-time fault diagnosis [24]. These studies demonstrate that knowledge distillation and quantization techniques offer practical solutions for reconciling the constraints of edge device resources with the demands for high performance, laying a solid foundation for the intelligent, real-time fault diagnosis of servo motors.

The studies above illustrate the application of deep learning in motor fault diagnosis, the evolution of edge computing in industrial contexts, and the advancements of knowledge distillation and model quantization in edge intelligence. These technologies have provided novel solutions for industrial monitoring and fault diagnosis in edge environments, significantly enhancing diagnostic efficiency and real-time performance. However, existing research still faces challenges, such as the difficulty of efficiently deploying complex deep learning models on edge devices, insufficient model generalization capabilities in multi-fault scenarios, and the need to further reduce computational demands in resource-constrained settings [25,26,27]. To address these challenges, researchers have proposed lightweight methods based on knowledge distillation, such as the self-supervised diversity knowledge distillation framework (SSD-KD), which effectively enhances the performance of lightweight models in classification tasks by integrating the instance relationship knowledge while significantly reducing the parameter count and computational requirements [28]. Additionally, through instance interaction and attribute-aware strategies, the dynamic interaction learning (DIL) framework achieves an outstanding performance for lightweight remote sensing detection models in complex, multi-scale scenarios [29].

On the other hand, researchers have also introduced lightweight solutions involving model quantization, such as the guided hybrid quantization and self-teaching framework (GHOST), which realizes the efficient remote sensing object detection on resource-constrained devices through multi-level quantization and self-teaching strategies [30]. The Adaptive reinforced supervision distillation (ARSD) framework combines multi-scale core feature imitation with rigorous regression distillation modules, significantly enhancing the accuracy and robustness of lightweight models [31]. This study proposes a lightweight diagnostic framework combining knowledge distillation and model quantization to advance the application of servo motor fault diagnosis technology in real-time monitoring and intelligent diagnosis on edge devices.

## 3. CNC System Architecture and Optimization Methods Based on Edge Intelligence

### 3.1. Overall System Architecture

The servo motor fault diagnosis system, set against the backdrop of intelligent manufacturing, aims to achieve the efficient real-time monitoring and diagnosis through edge intelligence. Unlike traditional cloud computing models, edge intelligence processes data near its source, effectively reducing latency and enhancing the real-time performance and synchronization in data processing. This paper presents a comprehensive architecture for an edge intelligence fault diagnosis system, as illustrated in Figure 1. The system is divided into the physical layer, cyber layer, and application layer, with each layer employing a hierarchical and modular approach to enhance system flexibility and processing efficiency, thereby providing robust technical support for servo motor fault diagnosis in intelligent manufacturing environments.

**Physical layer:** The physical layer comprises the actual servo motors and their associated sensors, including vibration sensors, temperature sensors, and others. These sensors capture real-time data on motor operational states, such as vibration amplitude and temperature fluctuations. These data are transmitted through edge nodes to the upper layers for processing. The primary role of the physical layer is to ensure comprehensive and real-time data acquisition, thereby providing accurate data input for fault diagnosis and condition monitoring.

**Cyber Layer:** The cyber layer represents the core computational component of the system, consisting of multiple edge nodes and edge servers dedicated to real-time data processing and intelligent diagnosis. Each edge node receives sensor data from the physical layer and conducts preliminary processing and analysis, including data cleansing, noise filtering, and feature extraction. The edge servers subsequently run intelligent algorithm modules, such as the MSCNN-LSTM-attention model, to perform the inference process for fault diagnosis. Moreover, the cyber layer includes functionalities like model training and parameter optimization, utilizing knowledge distillation and model quantization to make complex deep-learning models lightweight and compatible with the computational capabilities of edge devices. This architectural design ensures efficient and real-time data processing, supporting rapid motor fault diagnosis.

**Application layer:** The application layer provides users with a visualization interface and data analysis tools for motor fault diagnosis. Through the interactive interface, users can access real-time status data, historical records, and the diagnostic analysis reports of the motors. This layer integrates the computational outcomes of the cyber layer to enable the intuitive visualization of equipment operational status and real-time feedback, offering intelligent decision support and remote monitoring capabilities. The application layer effectively combines the immediate diagnostic capabilities of edge computing with the in-depth analytical power of cloud computing, providing robust technical support and operational optimization for the entire intelligent manufacturing system.

### 3.2. Information Model Structure of Edge-Intelligent CNC System

Figure 2 illustrates the information processing framework of a CNC system empowered by edge intelligence, encompassing the entire process from data acquisition and processing to final application and service. This architecture comprises the data access, processing, storage, middleware, and application layers, aiming to achieve intelligent, real-time, and efficient data management and control for CNC systems.

The data access layer connects with CNC systems, sensors, servo systems, and other equipment via industrial protocols (e.g., Modbus, OPC UA, Siemens S7) to enable the real-time collection of machine operation status and machining process data. Simultaneously, IoT protocols (such as MQTT, HTTP/TCP, and WebSocket) are employed to support data transmission from diverse sensing devices and systems, ensuring data collection’s comprehensiveness and real-time nature. Through the coordinated work of load balancing and data acquisition middleware, this layer guarantees that information from various data sources is efficiently relayed to the data processing layer.

The data processing layer forms the system’s core, consisting of edge access gateways, data parsing modules, data processing modules, and control logic. The edge access gateway handles the integration of different data streams and optimizes data transmission through load-balancing mechanisms. During the data parsing phase, the data fusion module integrates multi-source data and stores these in the data storage module for subsequent analysis and application. The data processing module primarily manages CNC system configuration and core equipment data and ensures data security. Following data processing, the application algorithms and control logic modules conduct further analysis of the processed results, ensuring the accuracy and timeliness of diagnosis and control. Additionally, this layer includes a rule engine and algorithm models for executing complex control strategies and task scheduling.

The storage and middleware layer is primarily responsible for data storage and management, encompassing various types of data storage units, such as statistical, real-time, and historical data. It comprehensively records event information, commands and responses, fault data, and more, facilitating subsequent analysis and operational traceability. This layer’s design endows the system with considerable flexibility and scalability regarding data security, management, and traceability.

The system offers modules for visualization applications, external service interfaces, and open APIs in the application layer, facilitating the interaction between end users and external systems. Visualization applications enable users to intuitively monitor the CNC system’s operational status, production efficiency, and diagnostic information, aiding decision-makers in making prompt and informed judgments. Furthermore, through external service interfaces and open APIs, the system seamlessly integrates with third-party systems, supporting intelligent control and management within industrial IoT environments.

This modular architecture design leverages the features of edge computing and IoT, allowing data processing and control to be performed in proximity to the data source, thereby significantly reducing data transmission latency and enhancing system responsiveness and robustness. Through the collaborative operation of edge access gateways and load-balancing mechanisms, the system achieves efficient data acquisition and processing capabilities, effectively supporting the intelligent operation of CNC systems and demonstrating the vital role of edge intelligence in modern industrial applications.

### 3.3. Lightweight Fault Diagnosis Algorithm Based on Knowledge Distillation and Model Quantization

Figure 3 illustrates the overall servo motor fault diagnosis model compression and deployment adopted in this study, encompassing two critical steps—knowledge distillation and model quantization—which ultimately enable lightweight model deployment. Through the combination of knowledge distillation and quantization, a servo motor fault diagnosis system capable of efficient operation on edge devices has been constructed, offering a solution for the real-time diagnostic needs of industrial environments.

In the initial development of the fault diagnosis model, a large-scale integrated network was first constructed as the teacher model, designed for the high-precision classification of servo motor faults. Given the high complexity of the teacher model, with substantial computational demands and memory usage, it is unsuitable for direct deployment on resource-constrained edge devices. Thus, knowledge distillation was employed to compress and lighten the model by transferring the knowledge from the teacher to a smaller student model.

The essence of knowledge distillation lies in guiding the student model to learn the teacher model’s output probability distribution (soft labels), thereby equipping the student model with classification capabilities similar to the teacher’s. During this process, the teacher model generates probability distributions for unlabeled samples, precisely conveying the similarities and differences between categories. Compared to traditional complex labels, soft labels provide more information regarding the uncertainty and boundaries between classes, allowing the student model to acquire more nuanced knowledge during training and enhancing its generalization capability. The student model’s learning process involves loss function calculations based on distillation loss, which includes the soft targets output by the teacher model and the distillation loss of the student model’s output. The distillation loss is computed by measuring the divergence between the student model and the teacher model’s output probabilities, effectively enhancing the student model’s performance. Through this process, despite significantly reduced complexity, the student model can achieve classification accuracy close to that of the teacher model.

Upon the completion of knowledge distillation, model quantization techniques were further employed to reduce the student model’s storage and computational requirements, converting it into a TensorFlow Lite format to fit the resource-constrained edge computing environment. The quantization process compressed the weights and computations of the student model, making it more lightweight without significantly compromising performance. Compared to the original student model, the quantized model has dramatically reduced storage requirements and improved computational efficiency, allowing it to run efficiently on low-power edge devices such as the Raspberry Pi 4B. The figure presents the specific steps of model quantization, including model reconstruction and conversion, model evaluation, and ultimately generating the lightweight student model, ensuring accuracy and efficiency during edge device deployment.

The quantized lightweight model was integrated into the intelligent servo motor fault diagnosis application and deployed on edge devices to achieve the industrial equipment’s real-time monitoring and fault diagnosis. Within this application, the quantized student model performs inference through the TF Lite standard runtime, and, combined with a user-friendly graphical user interface (HMI), users can efficiently conduct data collection, fault detection, and the visualization of diagnostic results. The application processes the collected servo motor vibration signals in real time and outputs diagnostic results, enabling operators in industrial environments to quickly identify faults, reduce equipment downtime, and improve maintenance efficiency.

The overall architecture design, utilizing the two fundamental techniques of knowledge distillation and model quantization, transforms the model from a high-precision, large-scale network into a lightweight version deployable on edge devices. Knowledge distillation ensures that the student model closely approximates the performance of the teacher model. At the same time, quantization significantly reduces the computational complexity and memory usage, enabling rapid responsiveness on resource-limited edge devices. This fully demonstrates the application value of edge intelligence technologies in modern industrial scenarios.

## 4. Optimization and Deployment Methods of Deep Learning Models

### 4.1. Model Construction

The servo motor fault diagnosis model employs a combination of MSCNN, LSTM, and attention mechanisms to handle fault data in complex industrial environments. This section provides a detailed description of the model’s architecture and its underlying theoretical support. The model construction process is shown in Algorithm 1.

First, a multi-scale convolutional neural network (MSCNN) is used to extract multi-scale features from the input vibration signals. Let the input signal be represented by matrix *X*, with dimensions T×C, where *T* denotes the time steps and *C* represents the number of channels of the input signal. In MSCNN, convolutions are performed on the input signal with different kernel sizes to capture features at multiple scales:(1)$$H_{k}=\mathrm{ReLU}\left(X*W_{k}+b_{k}\right),\;k\in \left\{3,5\right\}$$
where $W_{k}$ and $b_{k}$ represent the convolution kernels and bias terms, respectively, and *k* denotes the kernel size (3 and 5 are used here for different scales of convolution operations). Convolution, followed by the ReLU activation function, yields feature maps at different scales. These features are then passed through a max-pooling layer to reduce dimensionality and mitigate noise:(2)$$M_{k}=\mathrm{MaxPooling}\left(H_{k},p\right),\;p=4$$
where *p* is the pooling size used to reduce feature dimensions and extract the most significant local information. The features from different convolution scales, $M_{1}$ and $M_{2}$, are ultimately concatenated along the channel dimension to form the fused feature representation:(3)$$C=\mathrm{Concat}(M_{1},M_{2})$$

By combining multi-scale convolutions, MSCNN effectively captures the input signal’s local and global features, aiding in identifying fault characteristics across different frequency components.

Next, the data extracted through multi-scale feature extraction are fed into the LSTM network. The LSTM network models long-term dependencies in time-series data, thereby better capturing the dynamic information within the vibration signals. The LSTM computations are expressed as:(4)$$h_{t},c_{t}=\mathrm{LSTM}\left(C_{t},h_{t-1},c_{t-1}\right)$$
where $h_{t}$ and $c_{t}$ represent the hidden state and cell state at time step *t*, and $C_{t}$ is the feature input extracted by multi-scale convolution at time step *t*.

Following the LSTM output, an attention mechanism is introduced to further enhance the model’s ability to focus on key features. The core idea of the attention mechanism is to emphasize the features at critical time points by weighted summation, thereby improving the diagnostic accuracy of the model. Given the LSTM output sequence $\left\{h_{1},h_{2},\cdots ,h_{T}\right\}$, the attention weights $\alpha_{t}$ are computed via self-attention:(5)$$\alpha_{t}=\frac{\exp(\mathrm{score}\left(h_{t},H\right))}{\sum_{t=1}^{T}\exp\left(\mathrm{score}\left(h_{t},H\right)\right)}$$
where $\mathrm{score}(h_{t},H)$ computes the similarity score between the current time step feature $h_{t}$ and the entire sequence, typically using the dot product. The final attention-weighted output is:(6)$$A=\sum_{t=1}^{T}\alpha_{t}\cdot h_{t}$$

The attention-weighted feature *A* represents the aggregated critical information within the time series, reflecting the outcome after focusing on significant time points.

Finally, the attention-weighted features *A* are passed through a fully connected layer to achieve the classification task:(7)$$\hat{y}=\mathrm{softmax}\left(W_{\mathrm{dense}}\cdot A+b_{\mathrm{dense}}\right)$$
where $W_{\mathrm{dense}}$ and $b_{\mathrm{dense}}$ are the parameters of the fully connected layer, and $\hat{y}$ represents the predicted output, indicating the probability distribution over different fault types.

As the effective combination of MSCNN, LSTM, and attention mechanisms, the MSCNN-LSTM-Attention model efficiently extracts and focuses on fault features, achieving the high-precision classification of multiple fault states from sensor data. The algorithm process, as outlined in Algorithm 1, primarily includes multi-scale convolutional feature extraction, the LSTM-based modeling of temporal dependencies, and focuses on critical features via attention mechanisms, followed by outputting fault classification results through the fully connected layer.
**Algorithm 1** Fault diagnosis model construction process**Input:***X*: Input vibration signal matrix of size T×C**Output:**y^: Predicted fault type probabilities**Steps:**1:Initialize the model, including MSCNN, LSTM, and attention layers2:**Multi-scale convolution:**3:**for** each filter size k∈{3,5} **do**4:        Compute convolution feature: Hk=ReLU(X∗Wk+bk)▹ See Equation (Equation 1)5:        Apply max pooling to reduce dimensions: Mk=MaxPooling(Hk,p)6:**end for**7:Concatenate features from different scales: C=Concat(M1,M2)8:**LSTM Layer:**9:Use LSTM to process concatenated features and obtain hidden states10:**Attention mechanism:**11:Compute attention weights and weighted output: *A*12:**Fully connected layer:**13:Compute final output: y^=softmax(Wdense·A+bdense)       ▹ See Equation (Equation 7)

### 4.2. Knowledge Distillation and Student Model Construction

This study employs knowledge distillation to optimize the servo motor fault diagnosis model. The process of knowledge distillation used in this work involves several stages: first, training the teacher model and using it to generate soft labels from the training data; subsequently, these soft labels guide the training of the student model, enabling it to approximate the teacher model’s performance.

1.Teacher Model and Soft Label Generation

The teacher model adopted in this study is a relatively complex deep-learning model designed to learn the mapping between input data and output labels. The teacher model’s predictions typically contain richer information related to the classes, which can help the student model better understand inter-class similarities. These predictions are referred to as “soft labels”. Let the training set be (X,Y), where *Y* are the discrete complex labels, and the teacher model’s output is a class probability vector $\hat{Y}=\mathrm{softmax}\left(f_{t}\left(X\right)\right)$, expressed as:(8)$${\hat{Y}}_{i}=\frac{\exp(z_{i}/T)}{\sum_{j=1}^{C}\exp\left(z_{j}/T\right)}$$
where $z_{i}$ is the raw output score for class *i*, and *T* is the temperature parameter controlling the smoothness of the probability distribution. The probability distribution becomes smoother when T>1, enriching the teacher model’s class information. The teacher model is well trained, and its generated soft labels are the target for student model training.

2.Student Model Design

To extract and model fault features, the student model uses a simplified combination of MSCNN, LSTM, and attention mechanisms. It comprises an input layer, convolutional layer, LSTM layer, attention layer, and a fully connected layer.

Assume that the input is a three-dimensional tensor $X\in R^{N\times T\times C}$, where *N* denotes the number of samples, *T* represents the time steps, and *C* is the number of input channels. The input layer employs a one-dimensional convolution operation to extract local time-series features:(9)$$H_{\mathrm{conv}}=\mathrm{ReLU}\left(X*W_{\mathrm{conv}}+b_{\mathrm{conv}}\right)$$
where $W_{\mathrm{conv}}$ and $b_{\mathrm{conv}}$ are the convolution kernel and bias terms. The convolved features $H_{\mathrm{conv}}$ are passed to the LSTM layer to capture temporal dependencies in the input signal. For the LSTM layer, the recursive relationship is given by:(10)$$h_{t},c_{t}=\mathrm{LSTM}\left(H_{\mathrm{conv}},h_{t-1},c_{t-1}\right)$$
where $h_{t}$ and $c_{t}$ denote the hidden state and cell state at time step *t*. Through the LSTM, the model captures the dynamic features of the input signal over time.

Subsequently, an attention mechanism is used to further enhance the model’s ability to focus on key time-step features. Given the LSTM output sequence $\left\{h_{1},h_{2},\cdots ,h_{T}\right\}$, the attention weights $\alpha_{t}$ are computed using a dot product:(11)$$\alpha_{t}=\frac{\exp(h_{t}^{T}H)}{\sum_{t=1}^{T}\exp\left(h_{t}^{T}H\right)}$$

The attention-weighted output is:(12)$$A=\sum_{t=1}^{T}\alpha_{t}h_{t}$$

This attention-weighted feature representation focuses on the most crucial information in the time series and is further passed to a fully connected layer for classification:(13)$$\hat{y}=\mathrm{softmax}\left(W_{\mathrm{dense}}\cdot A+b_{\mathrm{dense}}\right)$$
where $W_{\mathrm{dense}}$ and $b_{\mathrm{dense}}$ are the parameters of the fully connected layer, and $\hat{y}$ represents the output class probabilities.

3.Student Model Training

To ensure that the student model effectively learns from the teacher model, soft labels are used as the training target for the student model. Precisely, the student model’s loss function consists of two components: one part is the cross-entropy loss of the soft labels, encouraging the student model to approximate the teacher model’s output distribution; the other part is the cross-entropy loss of the complex labels, maintaining the classification capability of the model:(14)$$L=\lambda L_{\mathrm{soft}}+\left(1-\lambda \right)L_{\mathrm{hard}}$$
where $L_{\mathrm{soft}}$ represents the soft label loss, $L_{\mathrm{hard}}$ represents the hard label loss, and λ is the weight parameter that balances the two loss components. During training, the hyperparameter λ is adjusted to find the optimal balance for the student model, achieving a favorable trade-off between diagnostic accuracy and model complexity.

Through the design and training process of the student model, knowledge distillation enables the student model to effectively approximate the performance of the teacher model in complex industrial environments. The specific process is shown in Algorithm 2. This algorithm details using soft and hard labels for joint training to improve the student model’s learning efficacy and ensure diagnostic accuracy.
**Algorithm 2** Student model knowledge distillation process**Input:***X*: Training dataset*Y*: Hard labels*T*: Trained teacher modelλ: Weight parameter for balancing loss**Output:***S*: Trained student model**Steps:**1:Initialize student model *S*, including convolution, LSTM, and attention layers2:Set learning rate η, number of training epochs *N*, batch size *B*, temperature Ttemp3:**for** each epoch from 1 to *N* **do**4:      Shuffle training dataset *X*5:      Split *X* and *Y* into batches of size *B*6:      **for** each batch (x,y) in (X,Y) **do**7:            Forward pass through teacher model to get soft labels: Y^=softmax(ft(x)/Ttemp)  ▹ See Equation (Equation 8)8:            Forward pass through student model: y^S=S.forward(x)9:            Compute soft label loss: Lsoft=CrossEntropy(Y^,y^S)10:          Compute hard label loss: Lhard=CrossEntropy(y,y^S)11:          Compute total loss: L=λLsoft+(1−λ)Lhard                                                              ▹ See Equation (Equation 14)12:          Backpropagation: Compute gradients of *L*13:          Update student model weights using learning rate η14:    **end for**15:    Evaluate on validation dataset: Compute validation loss and accuracy16:    Save model weights if validation accuracy improves17:**end for**18:Return trained student model *S*

### 4.3. Model Quantization and Edge Deployment

To effectively deploy the servo motor fault diagnosis model onto edge devices, the quantization of the trained student model is necessary to reduce its size, decrease computational and storage overhead during inference, and adapt to the resource constraints of edge devices. This study utilizes TensorFlow Lite quantization techniques to achieve efficient edge deployment of the model.

1.Quantization Overview

Model quantization is a technique that reduces a model’s computational and storage demands by lowering the precision of its parameters. Traditional deep learning models often use 32-bit floating-point representations during training and inference, which can be resource-intensive for edge devices. During the quantization process, the 32-bit floating-point parameters are reduced to 8-bit integers, significantly decreasing the model size, accelerating the computation process, and reducing storage and bandwidth requirements. Assuming the original model’s parameter matrix is $W\in R^{m\times n}$, the quantized parameters are represented as an integer matrix $\hat{W}\in Z^{m\times n}$. The mathematical expression for the quantization process is:(15)$$\hat{W}=\mathrm{round}\left(\frac{W-\min(W)}{\Delta }\right)$$
where Δ represents the quantization interval, and $\hat{W}$ represents the target precision (e.g., 8-bit precision). This linear transformation maps the floating-point matrix to a smaller numerical space, reducing storage needs.

2.TensorFlow Lite Quantization Implementation

In this study, the trained student model is quantized using the TensorFlow Lite Converter, generating a quantized model suitable for edge devices. TensorFlow Lite provides multiple quantization strategies; this work employs full quantization to reduce the model size while significantly maintaining inference accuracy. The quantization process involves the following steps:**Model conversion:** First, the trained student model f(x;W) is loaded and then converted using the TensorFlow Lite Converter. The model’s weight parameters are quantized to integer space:
(16)$$\hat{f}\left(x\right)=\mathrm{TFLiteConverter}\left(f\left(x;W\right)\right)$$**Quantization strategy:** During model quantization, TensorFlow Lite provides default optimization options, primarily by reducing the bit width of floating-point numbers to lower the precision restoration:
(17)$$\hat{W}=\arg\min_{W}{∥f\left(x;W\right)-\hat{f}\left(x;\hat{W}\right)∥}_{l}$$The objective is to ensure that the quantized model output closely approximates the original model, thus maintaining accuracy while reducing precision.**Generate quantized model:** After quantization, a TensorFlow Lite formatted quantized model is obtained, suitable for edge devices, significantly reducing memory usage and enhancing computational efficiency:
(18)TFLiteModel=Quantization(StudentModel)By enabling default optimization during quantization, the inference accuracy of the model remains as close as possible to the original while reducing memory requirements. The quantized model can then be deployed to edge devices, leveraging accelerated computation for inference. Finally, the quantized model is saved as a .tflite file for deployment on edge devices.

3.Edge Deployment

Deploying the quantized TensorFlow Lite model on edge devices enables the efficient inference and real-time diagnosis. Assuming that the input is *x*, the inference can be expressed as:(19)$$\hat{y}=f\left(x;\hat{W}\right)$$

During quantized inference, the parameters W^ are computed as 8-bit integers, significantly reducing the computational complexity and memory storage requirements. The complete model quantization and edge deployment process is detailed in Algorithm 3.
**Algorithm 3** Model quantization and edge deployment process**Input:***S*: Trained student model**Output:**S^: Quantized model ready for edge deployment**Steps:**1:Load the trained student model *S*2:Initialize TensorFlow Lite Converter with student model *S*3:**Quantization process:**4:Convert model weights from 32 to bit floating point to 8-bit integer: W^=roundW−min(W)Δ▹ See Equation (Equation 15)5:Optimize model using TensorFlow Lite optimization strategies6:**Generate quantized model:**7:Convert to TensorFlow Lite format: S^=TFLiteConverter(S)                                                                   ▹ See Equation (Equation 16)8:**Edge deployment:**9:Save the quantized model as student_model_quantized.tflite10:Deploy quantized model S^ to edge device11:Perform inference on edge device with reduced memory and computation requirements              ▹ See Equation (Equation 19)12:Return quantized model S^

## 5. Experimental Design and Validation

In exploring fault diagnosis for servo motors, the diversity of fault types and the demand for real-time performance in industrial environments are unavoidable challenges, particularly in the context of efficient production in intelligent manufacturing. Though effective under certain conditions, traditional methods based on manual feature extraction and expert knowledge must be revised to address more complex scenarios of industrial fault diagnosis. The shortcomings of these approaches become increasingly evident when faced with concurrent multi-fault types and dynamic changes in complex signals, highlighting the need for more advanced, deep learning-based diagnostic strategies. Deep learning methods can effectively capture the intricate nonlinear characteristics within signals, thereby enabling high-precision fault identification and enhancing production stability and efficiency through real-time inference capabilities.

To meet this challenge and improve diagnostic performance, this study adopts a deep learning strategy that integrates MSCNN, LSTM, and attention mechanisms. The model effectively learns the complex dynamic characteristics of servo motor operations by combining various feature extraction techniques, thereby achieving accurate and robust fault diagnosis.

Furthermore, we deployed this lightweight deep learning model in an edge computing environment to optimize fault diagnosis’s real-time performance and resource utilization. Through knowledge distillation and model quantization, we successfully achieved efficient deployment on edge devices such as the Raspberry Pi, conducting fault diagnosis validation via effective data collection and processing strategies. The experimental framework verified the efficacy of the deep learning model in enhancing diagnostic accuracy and efficiency. It demonstrated its feasibility for real-time monitoring on edge devices, providing significant support for adopting this technology in practical industrial applications.

### 5.1. Construction of Experimental Platform

The experiment utilized a servo motor fault diagnosis system, with the edge node implemented using a Raspberry Pi 4B, primarily for model deployment and real-time inference tasks at the edge. The quantized student model was deployed on the edge node, leveraging TensorFlow Lite’s lightweight inference engine to achieve efficient motor fault diagnosis. The edge node receives sensor data, performs preprocessing and inference, and transmits results to the application layer for visualization. The experiment’s data acquisition and edge processing are illustrated in Figure 4. The cloud control and computation node employed a deep learning server with an NVIDIA RTX 4090 GPU (NVIDIA Corporation, Santa Clara, CA, USA) to simulate the environment for training complex deep learning models. The knowledge distillation and model quantization processes were completed in the cloud, with the teacher model being comprehensively trained on a high-performance computing platform, followed by the generation of a lightweight student model suitable for the resource-constrained edge devices. Once cloud-based training and quantization were completed, the quantized model was exported and deployed to the Raspberry Pi 4B device via the network.

The experimental platform adopted a cloud-edge collaborative architecture, where the cloud server was mainly responsible for training the teacher model and generating the student model. At the same time, the edge node handled the inference and real-time monitoring of the quantized model. The experimental environment validated the feasibility of running a lightweight deep learning model on an edge node and demonstrated the diagnostic performance on a resource-limited device using the Raspberry Pi 4B. Through this cloud-edge collaborative approach, the system achieved real-time monitoring and efficient fault diagnosis of the servo motor, providing an effective solution for fault monitoring in intelligent manufacturing environments.

### 5.2. Experimental Procedure

In this section, we describe the experimental procedure to demonstrate the complete process from model training and quantization to edge deployment. The experiment has several stages, each with specific tasks and objectives.

#### 5.2.1. Teacher Model Training

In this experiment, a teacher model was constructed for servo motor fault diagnosis. This teacher model was the reference for subsequent knowledge distillation and edge deployment, providing high-precision performance. The training process included data preparation, model construction and training, and performance evaluation.

1.Data Preparation

Motor fault diagnosis is a critical technology for ensuring the stable operation of industrial equipment and enhancing production efficiency. However, in practical applications, motor systems often encounter mechanical and electrical faults, where data acquisition and processing accuracy directly impact the diagnostic outcomes. Most diagnostic systems currently rely on data collected under laboratory conditions, which, while ensuring data quality, often lack alignment with real industrial environments. Furthermore, traditional data acquisition systems lack standardized interfaces, resulting in poor data transmission and processing interoperability, making it challenging for them to adapt to diverse production systems. To address these issues, this study proposes a data acquisition system based on edge computing devices, integrating sensor technology with advanced data processing algorithms to achieve efficient fault acquisition and diagnosis for servo motors.

This study draws upon multiple references on edge devices [32,33] and ultimately selects the Raspberry Pi 4B as the core edge computing platform for the data acquisition system, paired with the MCC118 voltage measurement board to enable the high-precision acquisition of motor vibration and voltage signals. The Raspberry Pi 4B offers an optimal balance of cost-effectiveness, hardware flexibility, and edge computing capabilities, facilitating real-time preprocessing and transmission of collected data, thereby significantly enhancing system responsiveness and processing efficiency. To address the data volume challenges posed by high sampling frequencies, the system implements feature extraction functions on the Raspberry Pi, such as root mean square (RMS) values and frequency spectrum features (FFT), reducing transmission overhead and improving data utility. An accelerometer (unit: mV/g) is connected to the MCC118 voltage measurement board, capturing subtle vibration changes indicative of motor faults with precision. The MCC118’s multi-channel synchronous sampling capabilities ensure the temporal consistency and measurement accuracy of vibration signals, meeting the experimental data sampling frequency and precision requirements.

The system seamlessly integrates the Raspberry Pi 4B with the sensors through Python scripts, managing data acquisition, normalization, and time-series preprocessing tasks. The raw signals collected are normalized and segmented into time slices of 1024 data points via Python scripts to meet the input requirements of the fault diagnosis model. Python’s versatility further supports automated data processing and management. The system facilitates the transmission of feature data to the edge server using the OPC UA protocol, ensuring compatibility with industrial standards, while large volumes of raw data can be locally stored in batches. This study collected six categories of servo motor fault data, including broken bars, eccentricity, inter-turn short circuits, bearing inner race faults, bearing outer race faults, and standard operation data. Each dataset was normalized using MinMaxScaler to ensure that all feature dimensions fall within comparable numerical ranges, thereby preventing the training instability caused by significant feature magnitude differences. The normalized data were segmented into 1024-length time slices, with 1000 samples generated for each fault category. All samples were randomly shuffled to ensure independent and identically distributed data, minimizing bias during model training.

This study utilizes CSV files during the experimental phase for data storage, offering simplicity and ease of use, particularly that suitable for batch data processing requirements. The acquired acceleration and voltage signals are recorded in time series, ensuring seamless preparation for subsequent model training and validation. Additionally, the system supports data transfer to an Oracle database via a network interface, enabling long-term data management and efficient access while ensuring security and scalability. Feature data transmitted through the OPC UA protocol are stored in designated feature tables within the database, whereas raw time-series data can be selectively compressed and archived to optimize storage resource utilization. The dataset is partitioned into training and testing sets in an 80/20 ratio, ensuring that the model is trained on a comprehensive sample set and its generalization performance is rigorously validated. This structured approach provides a solid foundation for subsequent analytical processes.

This study employs an accelerometer to capture motor vibration signals and utilizes the DAQHAT function library to achieve precise voltage signal acquisition and linear conversion. The system ensures sufficient sampling density to capture critical features and high-frequency components within the motor vibration signals, providing reliable data support for the fault diagnosis model.

A comprehensive data acquisition framework has been designed to visually illustrate the system architecture and workflow, encompassing a motor fault simulation platform, an accelerometer, a Raspberry Pi 4B, an MCC118 voltage measurement board, and a data storage module. The accelerometer is mounted on the motor housing to capture analog signals transmitted to the Raspberry Pi 4B via the MCC118 measurement board. The Raspberry Pi processes the data using Python scripts for normalization and segmentation, storing the time-series data as CSV files. Key feature data are transmitted to the edge server via the OPC UA protocol and ultimately stored in an Oracle database, forming a multi-layer data structure that supports long-term analysis and modeling. Figure 1 illustrates the connections between the system components and the data flow, showcasing the complete process from data acquisition to transmission and storage, thereby providing efficient and reliable technical support for motor fault diagnosis.

2.Measurement and Data Collection

The dataset utilized in this study comprises vibration signals collected under six typical motor operating conditions, encompassing regular operation and five fault states: broken bar fault, eccentric fault, inter-turn short circuit fault, inner race bearing fault, and outer race bearing fault. Vibration signals were captured using an accelerometer, with the measured acceleration signals routed through an MCC118 voltage measurement board. These signals were processed in real-time by a Raspberry Pi 4B and subsequently recorded as time-series data in the CSV format for model training and validation. All signals underwent normalization to ensure data consistency and efficiency before storage.

The vibration signals were sampled at a frequency of 12 kHz, allowing for the precise capture of high-frequency vibration characteristics during motor operation. This high-resolution data effectively highlights the distinct features arising from mechanical faults (e.g., bearing and rotor faults) and electromagnetic faults (e.g., inter-turn short circuits), as detailed in Table 1. These datasets provide a high-quality input foundation for developing fault diagnosis models and support the analysis of motor faults in both time and frequency domains.

The time-domain and frequency-domain characteristics of data under six typical motor operating conditions are illustrated. As shown in Table 2, under normal conditions, the vibration signal exhibits small and uniform amplitude, with spectral energy concentrated in the low-frequency region (Table 2a,b). For broken bar faults, the vibration signal demonstrates significant low-frequency fluctuations, with markedly enhanced low-frequency energy in the spectrum (Table 2c,d). Eccentric faults are characterized by periodic amplitude fluctuations in the vibration signal, with prominent low-frequency harmonic features in the spectrum (Table 2e,f). Signals from inter-turn short circuit faults display high-frequency, irregular fluctuations and a significant enhancement in the high-frequency spectrum (Table 2g,h). Inner race-bearing faults exhibit periodic impact characteristics in the signal, with sharp high-frequency peaks distinctly visible in the spectrum (Table 2i,j). Outer race-bearing faults present weaker yet unstable impact characteristics, with apparent high-frequency features and harmonics in the spectrum (Table 2k,l). These examples vividly highlight the distinctive time-domain and frequency-domain features of different fault conditions, providing high-quality training and testing data for the fault diagnosis model.

3.Model Construction

During model construction, the teacher model was built using an architecture that integrated MSCNN, LSTM, and attention mechanisms for the precise diagnosis of servo motor faults. The model input consisted of vibration signal segments with dimensions of (256, 4). First, the MSCNN module extracted multi-scale features from the signals, with the convolution operations of different kernel sizes enabling the network to effectively capture features at varying time windows, enhancing sensitivity to fault characteristics. Subsequently, the LSTM module processed the temporal features extracted by MSCNN, further capturing the temporal dependencies in the signals and fully utilizing the dynamic information in the vibration signals. Finally, the attention mechanism was introduced, allowing the model to focus on the most important features across all time steps, thereby improving the classification accuracy for critical faults. Through the combined use of MSCNN, LSTM, and the attention mechanism, the teacher model efficiently extracted meaningful features from complex vibration data, ensuring the accurate classification of servo motor faults.

4.Model Hyperparameter Settings

The Adam optimizer adjusted the gradients, adaptively enhancing the model convergence speed. The learning rate was set to 0.001 to balance the convergence speed and stability. The cross-entropy loss function was used to measure prediction bias in this multi-class task. A batch size of 64 helped accelerate the training process while maintaining stability. An early stopping mechanism preserved the model weights that performed best on the validation set to prevent overfitting. The detailed settings of the hyperparameters during training are provided in the Table 3 below:

5.Model Training and Configuration

The model was trained for 50 epochs with a batch size of 64, and validation performance was monitored to save the model weights that performed best, preventing overfitting. During training, the accuracy and loss curves of the training and validation sets were recorded, as shown in Figure 5 and Figure 6. The training and validation accuracy curves (Figure 5) indicate that the training accuracy increased rapidly during the initial stages, stabilizing after epoch 10. The validation accuracy remained high and eventually approached 1.0, indicating the model’s strong generalization ability without significant overfitting. The training and validation loss curves (Figure 6) showed considerable fluctuations initially, which is typical as the model searches for optimal parameters. However, both losses stabilized after epoch 10, verifying the model’s convergence and robustness.

6.Performance Evaluation

During testing, the trained teacher model was used to make predictions on the test set, and its performance was evaluated using a confusion matrix and t-SNE feature visualization. The confusion matrix (Figure 7) illustrates the model’s classification performance across different fault types, showing that the model achieved near-perfect classification accuracy across all fault categories. The diagonal elements represent correct classifications, while the off-diagonal elements denote misclassifications, which were nearly zero, indicating high accuracy in the servo motor fault diagnosis. Additionally, to gain a more intuitive understanding of the model’s feature extraction capability, the high-dimensional features were reduced and visualized using t-SNE (Figure 8). The t-SNE plot reveals distinct clusters for different fault types in the two-dimensional space, with homogeneous samples tightly grouped and heterogeneous samples separated. This demonstrates the teacher model’s ability to distinguish between different fault modes effectively. The confusion matrix and t-SNE visualization thoroughly validated the teacher model’s high classification accuracy and feature discrimination capabilities, laying a solid foundation for subsequent model lightweight and edge deployment.

#### 5.2.2. Knowledge Distillation and Student Model Training

In the experiment, the student model learned from the teacher model through knowledge distillation to achieve a more lightweight fault diagnosis solution. While maintaining diagnostic accuracy, the student model significantly reduced the computational complexity, facilitating efficient deployment on edge devices. The training process encompassed model construction and training, hyperparameter settings, performance evaluation, and comparative analysis with the teacher model.

1.Model Construction and Knowledge Distillation Process

The student model adopted a simplified convolutional LSTM network architecture, incorporating convolutional layers (Conv1D), LSTM layers, and an attention mechanism to achieve feature extraction, temporal dependency learning, and critical feature enhancement. Unlike the teacher model, the student model has fewer convolutional layers and LSTM units, aiming to reduce model complexity while retaining diagnostic performance. It is better suited for deployment in resource-constrained edge environments.

During knowledge distillation, the student model did not rely solely on the original training data’s ground truth labels but also learned from the soft labels generated by the teacher model. These soft labels provide rich information regarding inter-class similarities through probability distributions, enabling the student model to better understand the fuzzy boundaries between classes. Compared to direct training using complex labels, soft labels contain more knowledge from the teacher model, such as sample similarities and class uncertainties, thus enhancing the student model’s generalization capability. Despite its simplicity, this knowledge distillation approach enabled the student model to effectively learn the distinctions between complex features and achieve a classification accuracy close to that of the teacher model.

2.Hyperparameter Settings

The student model’s training hyperparameters were largely aligned with the teacher model’s but were adjusted to optimize the training process. The specific hyperparameters are presented in the Table 4 below:

The student model’s loss function consisted of soft and hard labels, with a weight parameter set to 0.5 to balance the contributions of the soft and hard labels to training. This approach ensured high diagnostic accuracy while significantly reducing the number of parameters, making the student model suitable for edge deployment.

3.Model Training and Configuration

The student model was trained using a configuration similar to the teacher model but with reduced complexity to accommodate edge environments. The training and validation accuracy and loss curves were recorded, as shown in Figure 9 and Figure 10.

The training and validation accuracy curves (Figure 9) depict the accuracy trend of the student model during training. The training accuracy rapidly increased initially, stabilizing after the 10th epoch, while the validation accuracy continued to grow throughout training, ultimately approaching 1.0. This indicates that the student model quickly learned the data patterns and demonstrated strong generalization. However, the student model exhibited slight fluctuations in certain epochs, possibly due to the reduced parameter count, making it less sensitive to some complex features than the teacher model.

The training and validation loss curves (Figure 10) reflect the convergence process of the model. Initially, the training and validation losses were high but gradually decreased and stabilized, effectively reducing classification errors and converging to an optimal solution. Compared to the teacher model, the student model’s loss curve exhibited minor fluctuations in the early stages but converged more quickly overall, demonstrating efficient learning. Furthermore, the smaller network size contributed to faster training speed than the teacher model.

4.Performance Evaluation and Comparative Analysis

To comprehensively evaluate the student model’s performance, we utilized a confusion matrix, t-SNE feature visualization, and a comparative analysis of the teacher and student models.

The confusion matrix (Figure 11) illustrates the classification performance of the student model on the test set. The results indicate that the student model performed consistently across the majority of categories, though there were a few misclassifications in the “broken bar fault” category, with an accuracy of 98%. Compared to the teacher model, the student model showed a slight decrease in discriminatory capability after network simplification, particularly in distinguishing fault types with similar features. Nonetheless, the overall classification performance of the student model remains commendable, meeting the accuracy requirements for practical applications in edge environments.

The t-SNE feature visualization (Figure 12) was used to analyze the feature extraction capabilities of the student model. By reducing high-dimensional features to two dimensions, the t-SNE plot illustrates the distribution of various samples in the feature space. The plot reveals that the student model effectively distributes different class samples within the feature space, with apparent clustering for similar samples. Compared to the teacher model, there is little difference in clustering quality. Although the clustering tightness is slightly lower for specific categories, the overall feature distribution still exhibits good separability, with distinct boundaries for most categories, meeting the requirements of practical applications. This indicates that, despite the simplification of network complexity, the student model’s feature extraction capability remains sufficient to handle the differentiation tasks of multiple fault categories.

The performance comparison analysis (Figure 13) illustrates the accuracy comparison between the teacher and student models on the test set. Although the teacher model showed a slight advantage in accuracy, the student model’s precision was comparable, thanks to the guidance provided by soft labels during the distillation process, which allowed the student model to leverage the teacher model’s knowledge to enhance classification performance. Additionally, the student model’s simplified structure significantly reduced the computational and storage requirements, making it suitable for real-time fault diagnosis on resource-limited edge devices, achieving the goals of efficient training and rapid inference.

5.Experimental Conclusion

The performance of the student model in servo motor fault diagnosis was comprehensively evaluated through the analysis of the training and validation accuracy and loss curves, confusion matrix, t-SNE feature visualization, and comparative analysis with the teacher model. Although the student model’s classification accuracy was slightly lower than the teacher model’s, its lightweight characteristics make it highly suitable for edge deployment, demonstrating strong practicality. The distilled model shows promise in maintaining high accuracy while achieving lightweight deployment, providing a feasible solution for the actual deployment of edge intelligence in the future.

#### 5.2.3. Quantization and Edge Deployment Analysis of the Student Model

This section discusses the performance of the student model after quantization on edge devices. By using TensorFlow Lite to generate the quantized model, we were able to significantly reduce its storage requirements and computational complexity, thus enabling efficient operation on resource-constrained edge devices. This experiment compared the performance of the original and quantized models in terms of inference time, memory usage, CPU utilization, and throughput. The following are the specific experimental results and analysis.

1.Model Quantization and Inference Time Comparison

As shown in Figure 14, the inference times of the original and quantized models were compared using a logarithmic scale to illustrate the differences better. The results indicate that the quantized model’s inference time is significantly better than the original model’s, primarily concentrated within the log range of −2 to −1.5. In contrast, the original model shows a longer and smoother distribution. This demonstrates that the quantized model substantially improves inference efficiency, with significantly reduced inference time, making it well suited for edge environments requiring rapid response.

2.Memory Usage Comparison

Figure 15 shows the memory usage of the original and quantized models during inference. It can be observed that the memory usage of the quantized model mainly remains constant, whereas the original model’s memory usage gradually increases during inference. Specifically, the average memory usage for the quantized model is 502.38 MB, while that of the original model is 501.67 MB. Although the memory usage of the quantized model is slightly higher, it exhibits minimal fluctuation and excellent stability. This stability makes the quantized model more suitable for running on edge devices, ensuring consistent resource utilization during extended operations.

3.CPU Utilization Comparison

Figure 16 depicts the CPU utilization of the original and quantized models during inference. The results show that the quantized model has virtually negligible CPU usage during inference, whereas the original model exhibits noticeable CPU usage peaks for some inference samples. This indicates that the quantized model utilizes CPU resources more efficiently during execution, avoiding substantial CPU load and making it more suitable for resource-limited applications on edge devices.

4.Advantages of the Quantized Model and Comprehensive Evaluation

To further compare the performance of the quantized and original models across different metrics, the following Table 5 summarizes the analysis.

The table shows that the quantized model demonstrates significant advantages across multiple aspects. The inference time is considerably reduced, and throughput is markedly increased, reaching nearly five times that of the original model. Although the memory usage of the quantized model is slightly higher, its minimal variation and stable resource usage make it advantageous. Additionally, the quantized model’s size decreased from 0.14 MB to 0.02 MB, providing substantial savings in storage and computational resources on edge devices.

Based on these comparative analyses, we can conclude that the quantized student model not only significantly outperforms the original model in inference speed and resource utilization, but also possesses lightweight characteristics that make it highly suitable for edge deployment, providing a reliable solution for real-time fault diagnosis.

#### 5.2.4. Robustness Analysis and Visualization Results Under Noisy Conditions

This section presents supplementary experiments and analyses of the model’s performance under various noise conditions. By testing both the teacher and student models under noise-free conditions (no noise) and with signal-to-noise ratios (SNRs) of 20 dB, 10 dB, and 5 dB, we compared their classification accuracy and confusion matrix results to evaluate their robustness in handling noise interference in practical industrial environments.

1.Accuracy Comparison Analysis

As illustrated in Figure 17, we compared the classification accuracy of the teacher and student models across four conditions: noise-free, 20 dB, 10 dB, and 5 dB. The results indicate that, as the SNR decreases, the recognition performance of both models deteriorates to varying extents. The teacher model achieves nearly 100% accuracy in noise-free conditions and maintains approximately 90% accuracy even at 5 dB, demonstrating strong robustness to noise. Comparatively, the student model shows a slight decline in accuracy under high-noise conditions but still achieves close to 87% accuracy at 5 dB. This indicates that, despite being lightweight, the student model retains a notable resilience against noise perturbations.

2.Confusion Matrix Analysis

To further analyze the impact of noise on fault diagnosis results at the class level, this section presents two representative confusion matrices. As shown in Figure 18, the teacher model demonstrates nearly error-free recognition across all fault categories under noise-free conditions, highlighting its exceptional performance in ideal signal environments. In contrast, Figure 19 illustrates the classification performance of the student model under 5 dB noise conditions. While the accuracy decreases compared to noise-free scenarios, the student model can still accurately identify most fault categories, with only a limited degree of misclassification observed between a few categories. This result visually underscores the interference caused by noise on feature extraction and classification decisions while also demonstrating the model’s commendable usability under lower SNR conditions.

3.Noise Experiment Evaluation

The results demonstrate that the proposed model framework retains a commendable level of robustness when subjected to varying degrees of noise interference. The high accuracy under noise-free conditions is a benchmark, showcasing the model’s exceptional performance with pristine signals. The model sustains a relatively high recognition rate even under the challenging 5 dB noise conditions. This trend can be attributed to the increased interference of noise with signal features at lower SNR levels, which significantly complicates feature extraction and classification processes. Particularly for fault types with similar patterns or subtle distinctions, noise exacerbates the difficulty of differentiation, leading to uncertainty near decision boundaries and consequently reducing classification accuracy.

Moreover, the teacher model demonstrates greater robustness under noisy conditions than the student model, highlighting the trade-off in performance incurred during the lightweight process of knowledge distillation. Nonetheless, the student model retains significant practical value for deployment on resource-constrained edge devices, effectively addressing complex environmental conditions in real industrial scenarios and providing a reliable solution for real-time fault diagnosis.

### 5.3. Deployment and Implementation of Fault Diagnosis Application

This section describes the deployment process and implementation methodology of the servo motor fault diagnosis application on edge devices. This application performs servo motor fault diagnosis using the quantized student model. It interacts with users through a user-friendly graphical user interface (GUI). It delivers an efficient, straightforward, intuitive operational experience suitable for on-site diagnostic needs within the Industrial Internet of Things (IIoT) environment.

1.Application Design and Implementation

The servo motor fault diagnosis application was developed using Python, with an intuitive graphical user interface designed based on the Tkinter library, as shown in Figure 20. The application aims to enable users to diagnose real-time motor faults with minimal effort. The interface design includes various functional buttons, such as “load sample” and “identify status,” which are delineated to provide an easy-to-use operational experience for on-site users.

The application utilizes the quantized student model to identify fault statuses for the servo motor. With its compact size and low computational requirements, the quantized model is well suited for operation on resource-constrained devices. Users can load motor vibration samples directly through the interface and view the real-time identification results. Additionally, the application integrates a training history feature that allows users to view accuracy and loss curves from the model’s training process, providing insights into the model’s learning performance. The entire design process prioritized operational convenience and user experience in industrial settings, aiming to simplify the complex diagnostic process, enabling users to complete the entire process—from data loading to fault identification—in just a few clicks.

2.Edge Device Deployment and Application

The fault diagnosis application was successfully deployed on the edge device Raspberry Pi 4B to achieve the goal of real-time monitoring and fault diagnosis on industrial sites. The student model embedded in the application underwent quantization, reducing its storage size from 0.14 MB to 0.02 MB. Thus, the model can operate effectively on the edge device, saving storage space and significantly reducing the computational overhead.

Deploying the student model on Raspberry Pi 4B equipped the application with real-time inference capabilities, enabling the rapid processing of collected motor vibration data and outputting fault diagnosis results. The application deployed on the edge device operates smoothly and has a responsive interface. Upon loading samples, the model delivers diagnostic results almost instantaneously, significantly enhancing the efficiency of use on industrial sites. This edge computing approach reduces dependence on a central server and improves response speed, providing a more reliable and autonomous solution for industrial environments.

3.Advantages and Practicality of the Application System

This application’s primary advantage lies in its edge deployment’s flexibility and efficiency. By quantizing the model to TensorFlow Lite format and deploying it on the edge node, the entire diagnostic system achieves full process automation—from sensor data acquisition to fault prediction. Compared with traditional cloud-based diagnostic solutions, the edge-deployed solution significantly reduces data transmission latency, avoiding potential diagnostic delays caused by network lag.

Edge deployment also enhances system robustness and privacy, as data need not be uploaded to the cloud for processing, which is particularly important in industrial scenarios with stringent privacy and security requirements. Moreover, the independent operation capability of edge nodes endows the application system with solid fault tolerance, ensuring continuous functionality even under unstable network conditions. This is critical for the continuous operation and maintenance of industrial equipment. With these advantages, this application demonstrates the broad potential for use in the IIoT, offering an effective solution for achieving more intelligent and efficient equipment fault management.

### 5.4. Summary of Algorithms and Comprehensive Evaluation

In this section, we summarize the various algorithms applied in the servo motor fault diagnosis process and comprehensively evaluate their performance in practical industrial scenarios.

By constructing teacher and student models, we employed several advanced deep learning techniques, including MSCNN, LSTM, attention mechanisms, and knowledge distillation, to achieve efficient and precise servo motor fault diagnosis. The teacher model, serving as a high-accuracy benchmark, adopted the MSCNN-LSTM-Attention architecture, which effectively extracts features across different temporal scales and captures temporal sequence information, using the attention mechanism to further focus on critical features, and thereby significantly enhancing the diagnostic accuracy. The student model learned from the teacher model through knowledge distillation, utilizing a simplified network architecture to deploy edge devices efficiently. Despite the structural simplification, the student model retained robust feature extraction capabilities through knowledge distillation, ensuring high diagnostic accuracy.

To evaluate the model performance, this study conducted a multifaceted assessment. Using confusion matrices and t-SNE dimensionality reduction visualization, we verified the teacher and student models’ classification ability and feature learning effectiveness across various fault types. The quantized student model was further deployed on a Raspberry Pi edge device, and the application provided a simple and intuitive graphical user interface for industrial on-site users, facilitating fault diagnosis even by non-expert operators. Experimental results demonstrated that the quantized model excelled in inference latency and memory usage, making it well suited for application in resource-constrained edge computing environments.

## 6. Summary and Outlook

### 6.1. Conclusions

In this study, we focused on developing an edge intelligence-based servo motor fault diagnosis system, comprehensively analyzing the model’s architecture, training process, deployment strategy, and practical application in industrial environments. Our main contribution lies in creating an efficient, lightweight servo motor fault diagnosis model that fully leverages edge intelligence technology, providing an innovative solution for real-time condition monitoring in IIoT environments.

We explored the growing demand for intelligent servo motor fault diagnosis. We proposed an edge intelligence-based solution to overcome the limitations of traditional centralized systems in terms of real-time responsiveness and resource constraints. To address these challenges, we developed a teacher–student model framework, where the teacher model served as a high-precision benchmark, and the student model was optimized through knowledge distillation to be a lightweight version suitable for edge environments. The system improved diagnostic accuracy and computational efficiency by integrating advanced deep-learning architectures.

By quantizing the student model and deploying it on edge devices like Raspberry Pi, we validated the feasibility of using lightweight neural networks for real-time inference on edge devices, significantly reducing inference latency and computational burden. This study provided a detailed evaluation of the performance of the quantized student model compared to its unquantized counterpart, including accuracy, inference time, memory usage, and robustness. Through a case study of deploying the fault diagnosis application on an edge device, we demonstrated the practical efficacy of this edge solution in industrial environments, enabling fast responses and reliable fault diagnosis capabilities.

Moreover, this study featured the design of a GUI-based application that provided a user-friendly interaction interface for on-site operators, simplifying the motor fault diagnosis process and allowing intuitive access to model training history and prediction results. The system can independently handle real-time fault analysis, reducing dependency on a central server and thus enhancing the operational reliability on industrial sites. This research demonstrates the potential of edge intelligence in extending the capabilities of intelligent maintenance systems, paving the way for its widespread application in the IIoT, and offering a practical solution for achieving brighter, more efficient equipment management.

### 6.2. Limitations and Future Work

Despite the significant achievements of this study, we acknowledge its limitations and outline several directions for future work.

**Expansion of theory and methods:** The current research is primarily confined to specific servo motor fault diagnosis systems and particular industrial application scenarios. Future work will extend the models and theories developed herein to a broader range of systems and environments to validate further and enhance their generalizability and practicality. Particularly in more complex industrial settings, exploring and verifying the model’s adaptability is essential.

**In-depth study of model and algorithm performance:** The discussion on the model and algorithm’s real-time performance, stability, and reliability requires further deepening. Future efforts will focus on optimizing these performance metrics to enhance the application value of the servo motor fault diagnosis system in the industrial domain. Specifically, under the resource constraints of edge devices, exploring ways to improve inference speed further and reduce memory consumption is a topic that warrants in-depth investigation.

**Large-scale experimental validation:** We plan to conduct experiments in a broader range of real-world industrial scenarios under diverse conditions to strengthen the practical applicability and reliability of the research outcomes, ensuring a close integration of theory and practice. This will involve testing the model’s performance on different devices and in varied operational environments to address the challenges that may arise during real-world deployment, thereby enabling the proposed solution to serve a more comprehensive array of industrial applications.

## Figures and Tables

**Figure 1 sensors-25-00009-f001:**
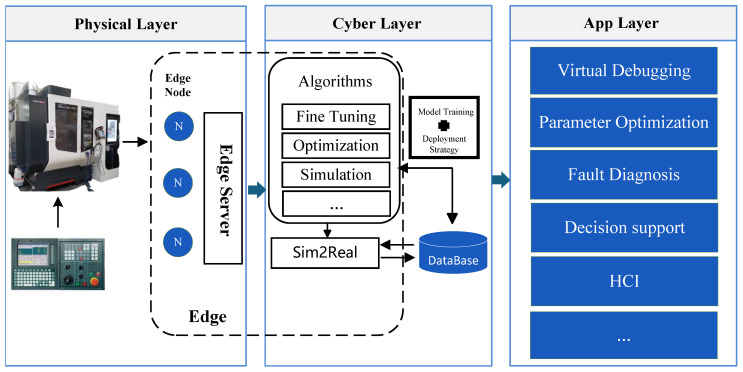
Overall system architecture.

**Figure 2 sensors-25-00009-f002:**
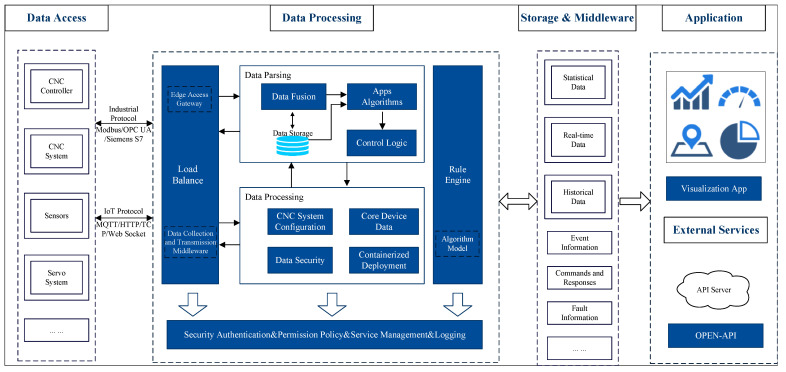
Information model structure of CNC system based on edge intelligence.

**Figure 3 sensors-25-00009-f003:**
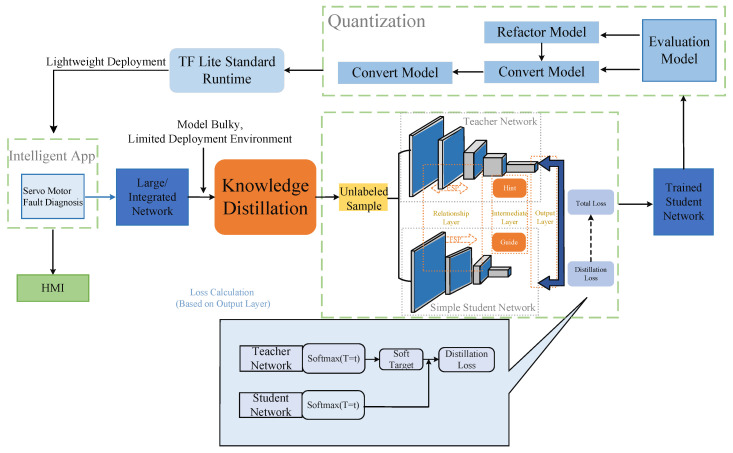
Diagram of lightweight fault diagnosis algorithm based on knowledge distillation and model quantization.

**Figure 4 sensors-25-00009-f004:**
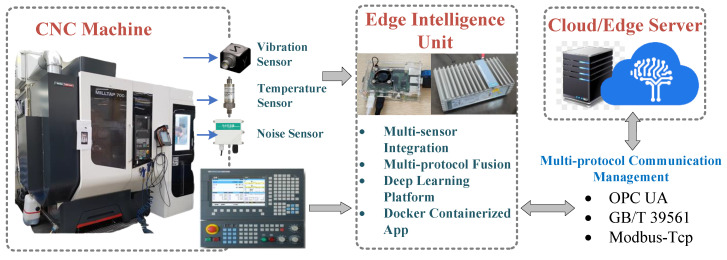
Experimental platform of edge-intelligent CNC system.

**Figure 5 sensors-25-00009-f005:**
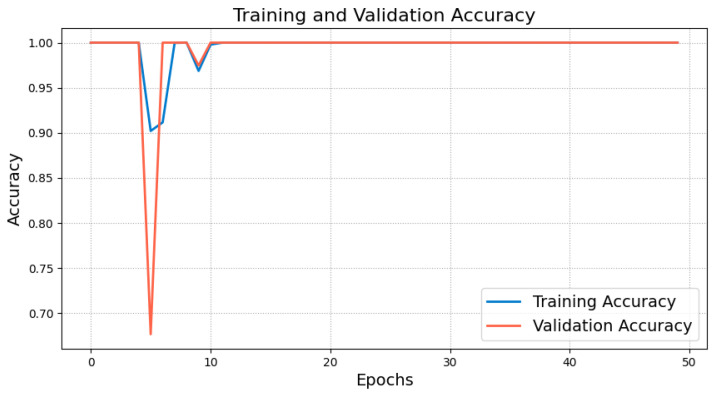
Training and validation accuracy curve of teacher model.

**Figure 6 sensors-25-00009-f006:**
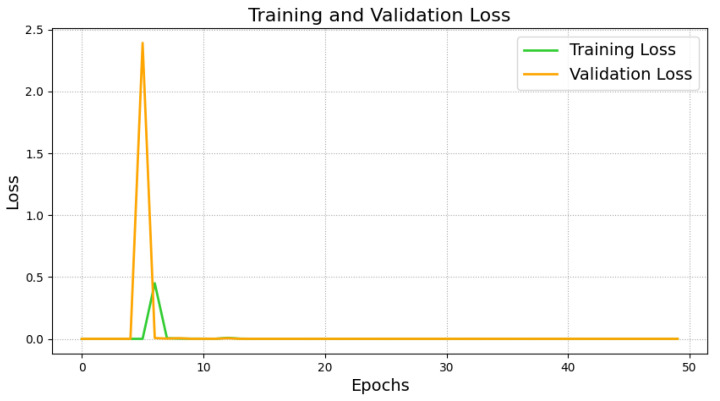
Training and validation loss of teacher model.

**Figure 7 sensors-25-00009-f007:**
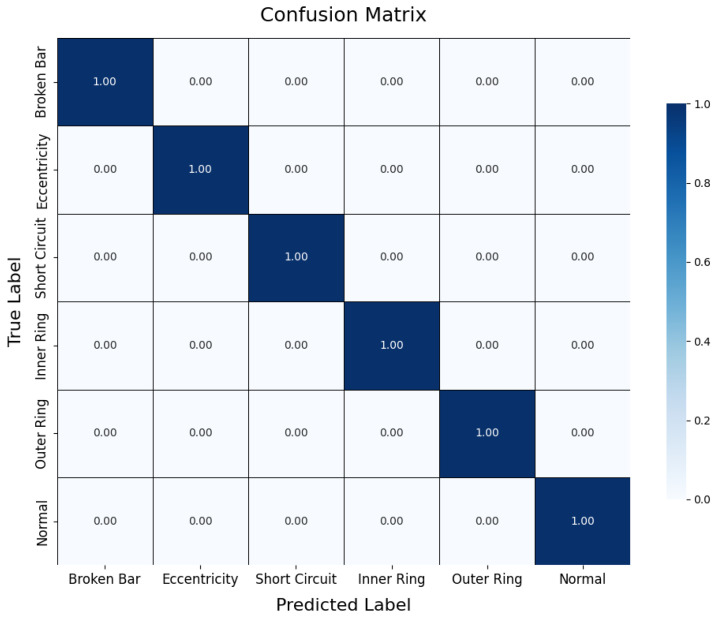
Confusion matrix of teacher model.

**Figure 8 sensors-25-00009-f008:**
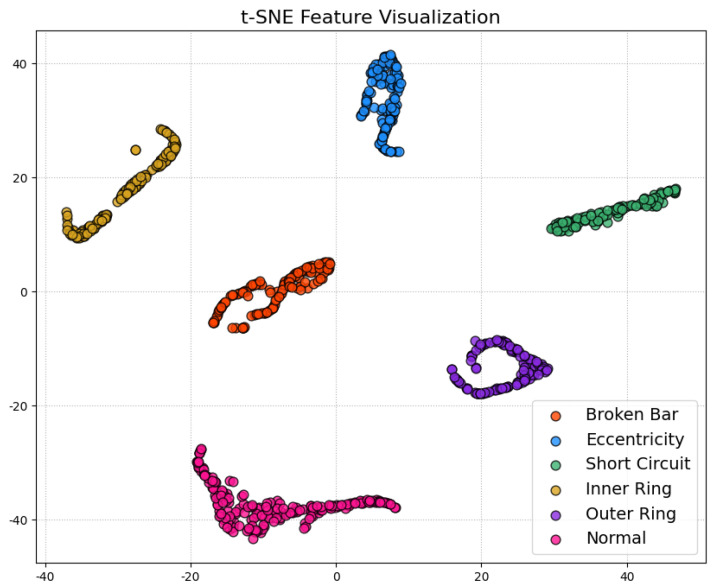
t-SNE feature visualization of teacher model.

**Figure 9 sensors-25-00009-f009:**
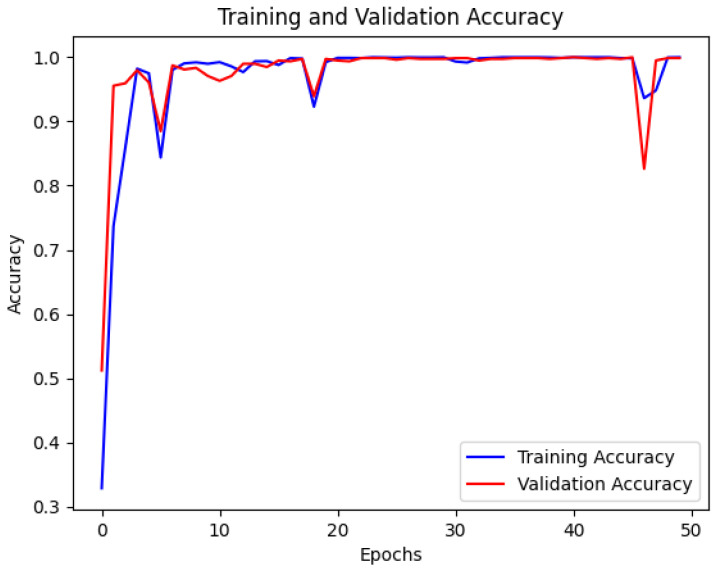
Training and validation accuracy curve of student model.

**Figure 10 sensors-25-00009-f010:**
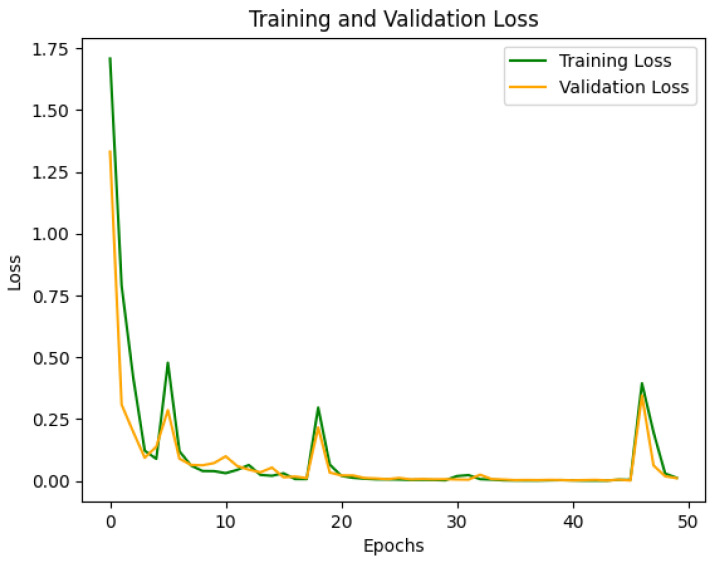
Training and validation loss curve of student model.

**Figure 11 sensors-25-00009-f011:**
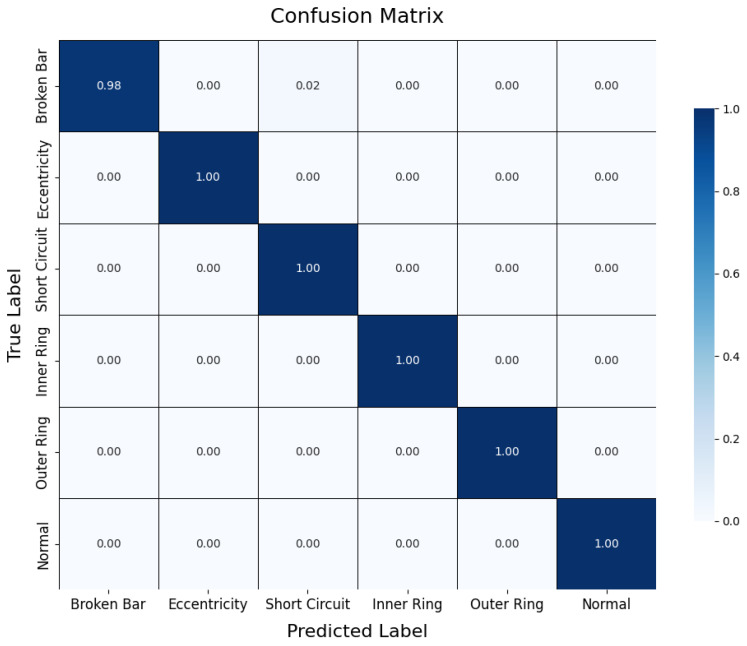
Confusion matrix of student model.

**Figure 12 sensors-25-00009-f012:**
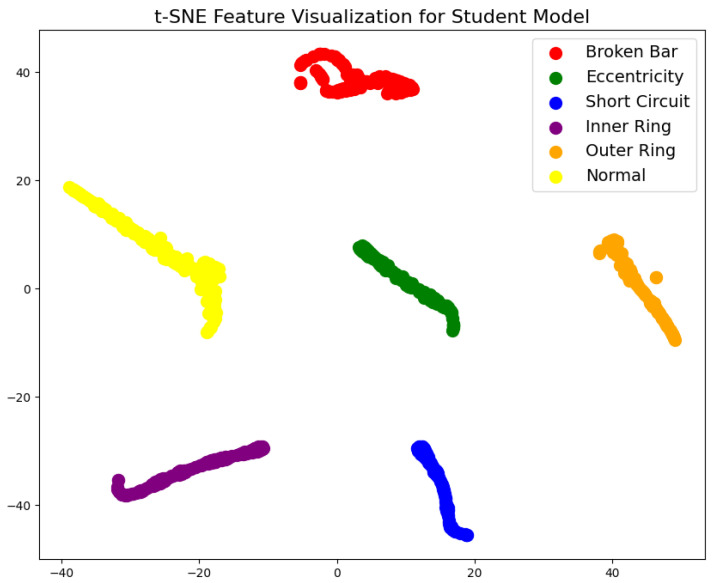
t-SNE feature visualization of student model.

**Figure 13 sensors-25-00009-f013:**
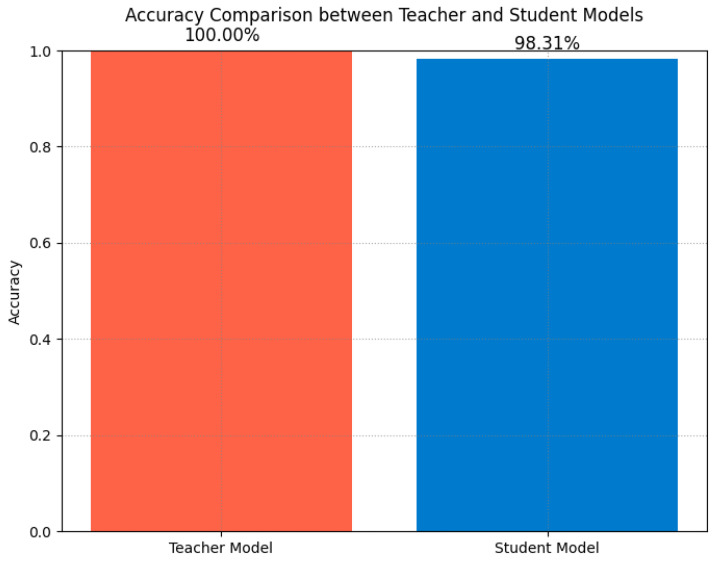
Accuracy comparison.

**Figure 14 sensors-25-00009-f014:**
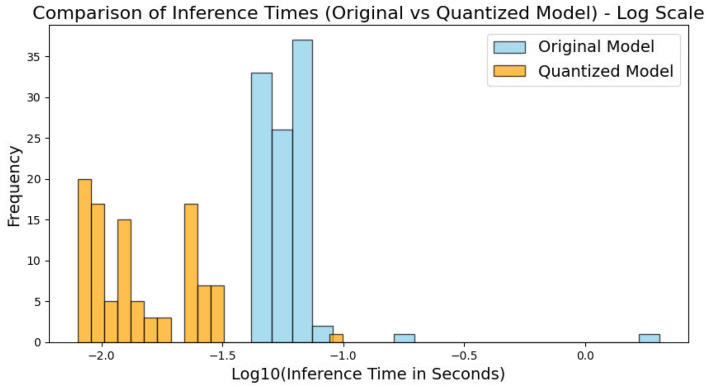
Comparison of inference times.

**Figure 15 sensors-25-00009-f015:**
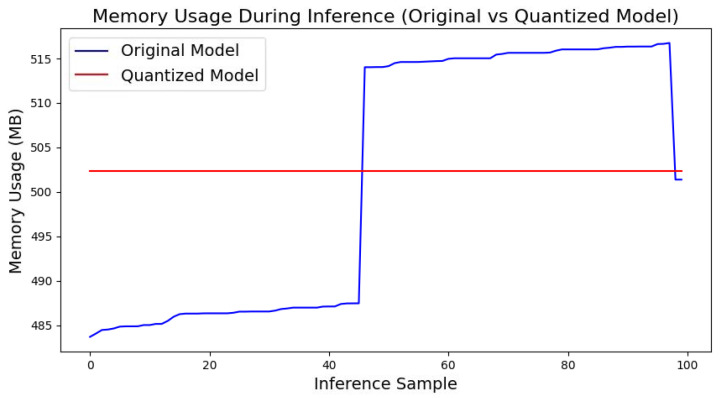
Memory usage during inference.

**Figure 16 sensors-25-00009-f016:**
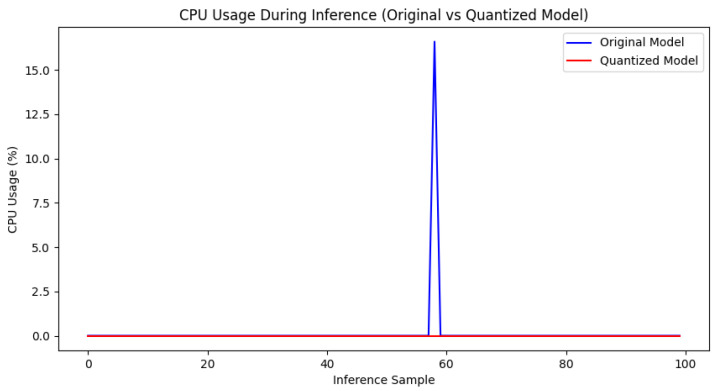
CPU usage during inference.

**Figure 17 sensors-25-00009-f017:**
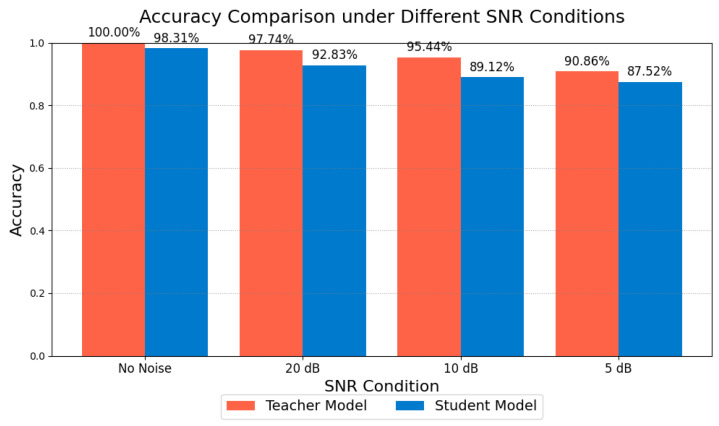
Accuracy of teacher and student models under varying SNRs.

**Figure 18 sensors-25-00009-f018:**
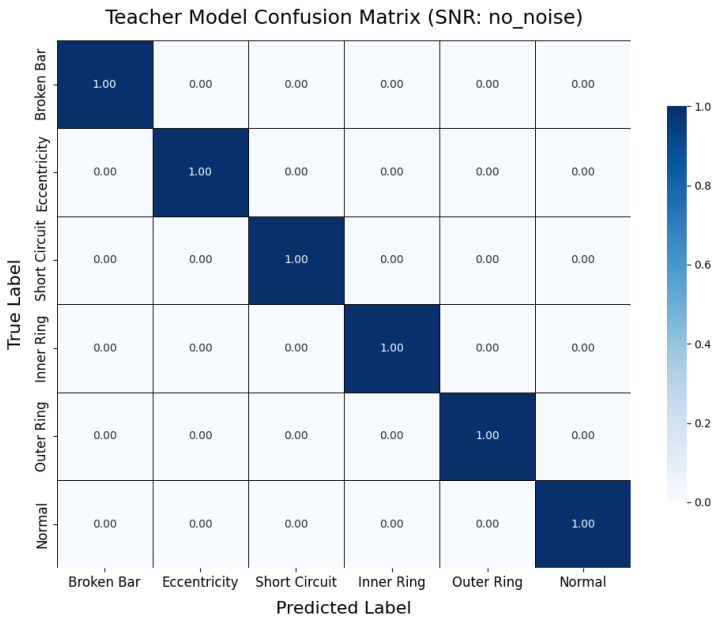
Teacher model confusion matrix (noise-free).

**Figure 19 sensors-25-00009-f019:**
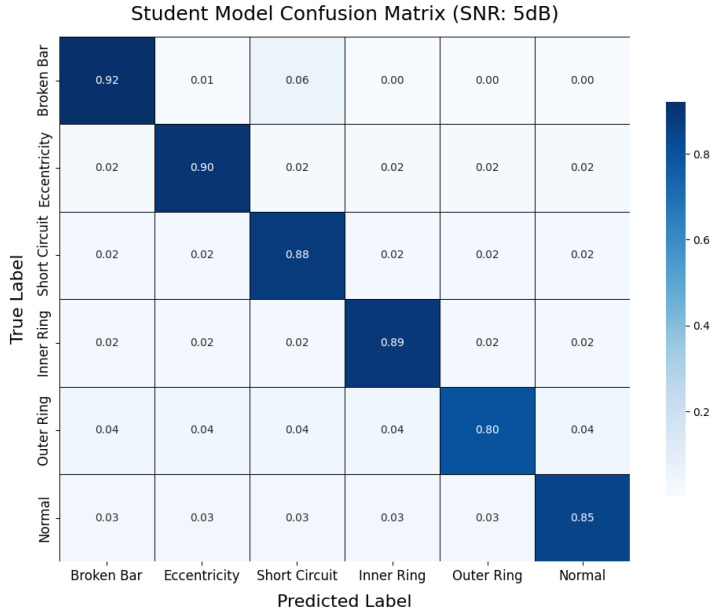
Student model confusion matrix (5 dB noise).

**Figure 20 sensors-25-00009-f020:**
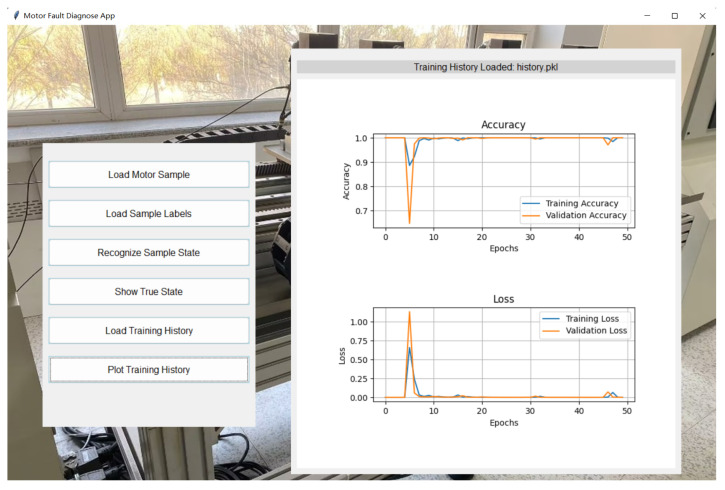
Servo motor fault diagnosis application.

**Table 1 sensors-25-00009-t001:** Fault types and their key characteristics.

Fault Type	Time-Domain Features	Frequency-Domain Features
Normal	Uniform amplitude, minimal fluctuation	Low-frequency focus, uniform high-frequency.
Broken bar	Significant low-frequency fluctuation	Enhanced low-frequency energy, weak harmonics
Eccentricity	Periodic fluctuation, uneven amplitude	Prominent low-frequency harmonics, distinct peaks
Short circuit	High-frequency, irregular fluctuation	Enhanced high-frequency energy, complex spectrum
Inner race fault	Strong impacts, consistent period	Sharp high-frequency peaks, clear harmonics
Outer race fault	Weaker impacts, unstable fluctuation	Prominent high-frequency peaks, weaker harmonics

* Typical characteristics of each fault type in time and frequency domains.

**Table 2 sensors-25-00009-t002:** Examples of Fault Data.

Type	Time Domain	Frequency Domain
Normal	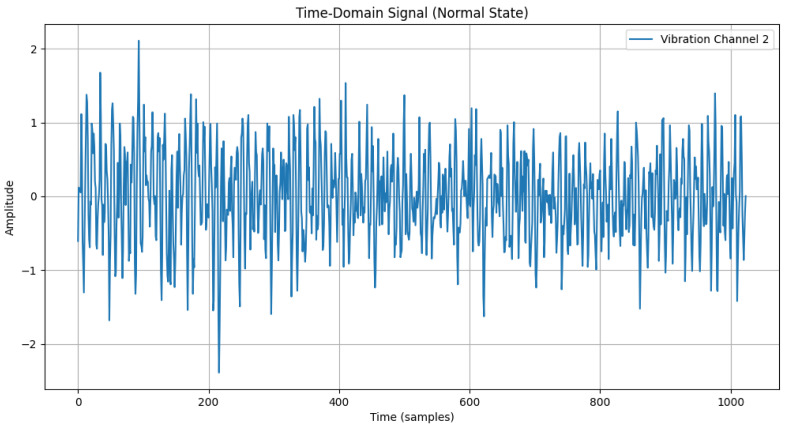 **a**	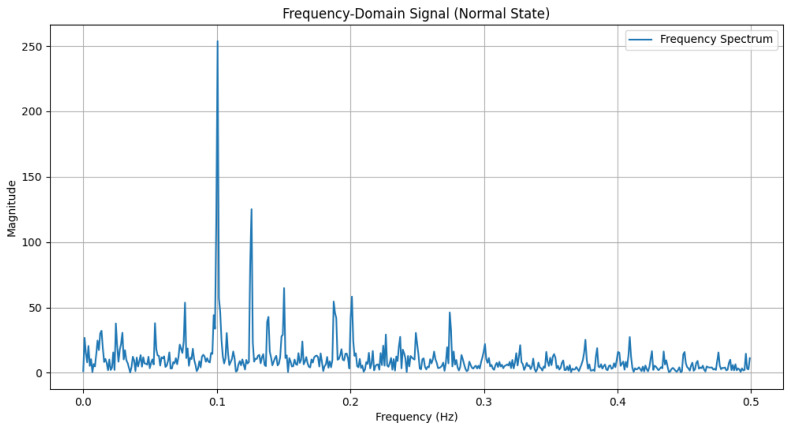 **b**
Broken bar fault	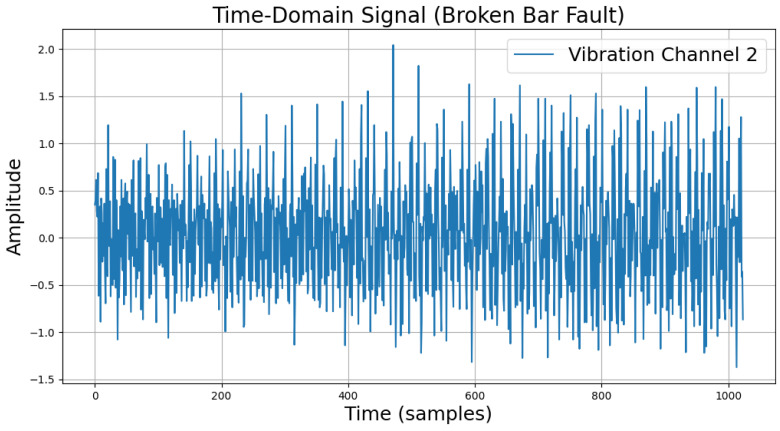 **c**	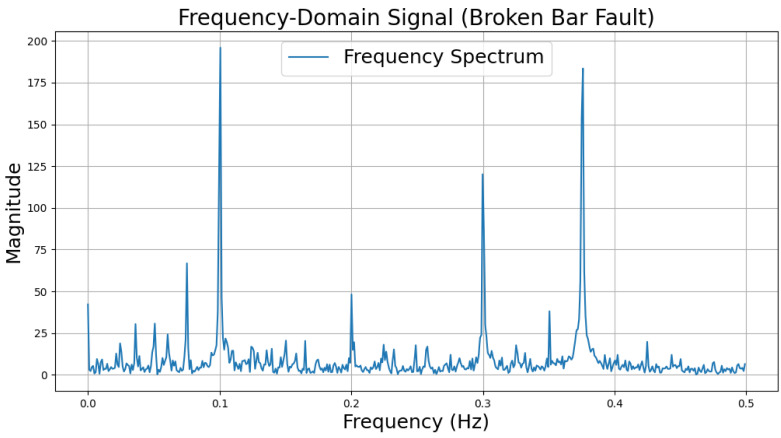 **d**
Eccentric fault	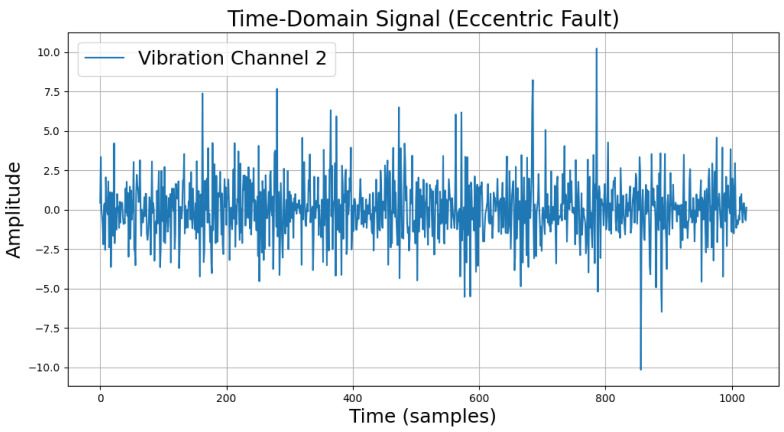 **e**	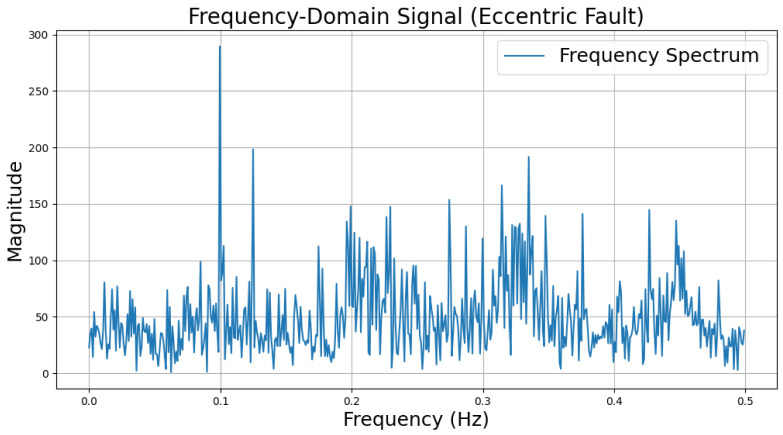 **f**
Inter-turn short circuit	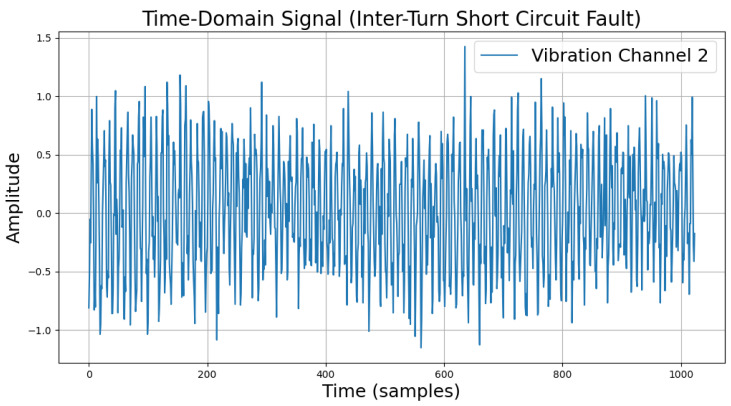 **g**	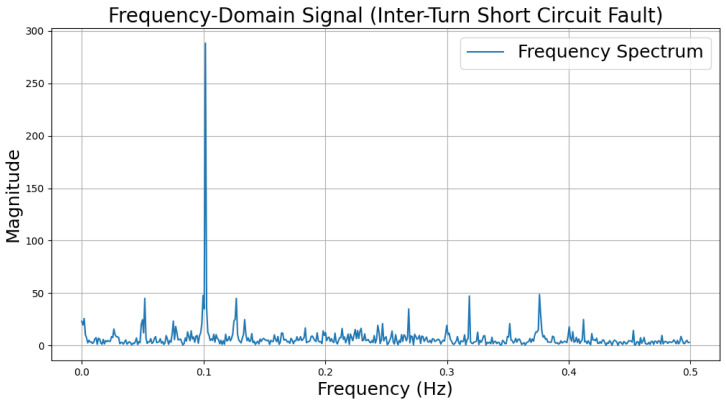 **h**
Inner race bearing fault	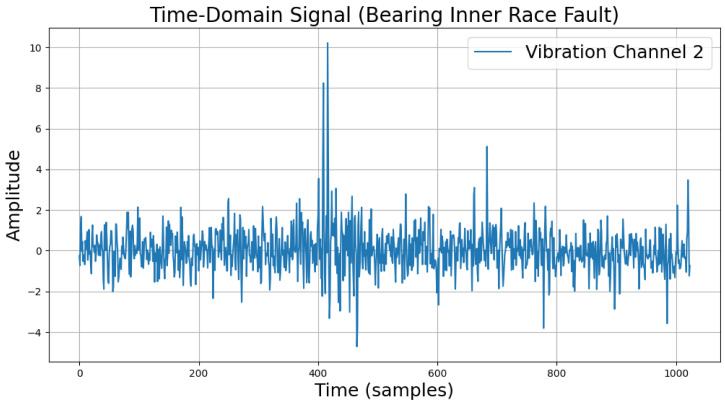 **i**	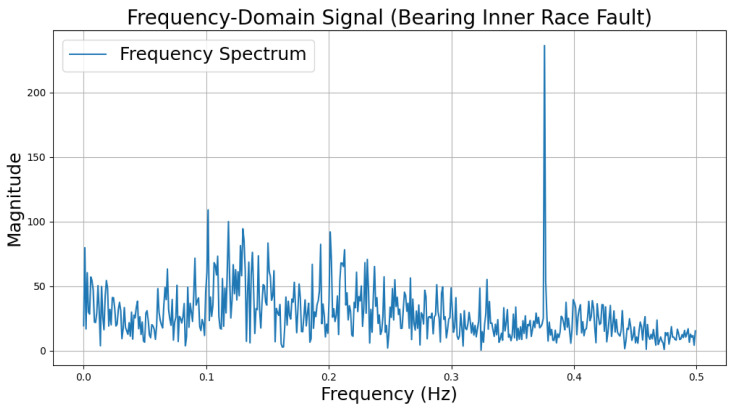 **j**
Outer race bearing fault	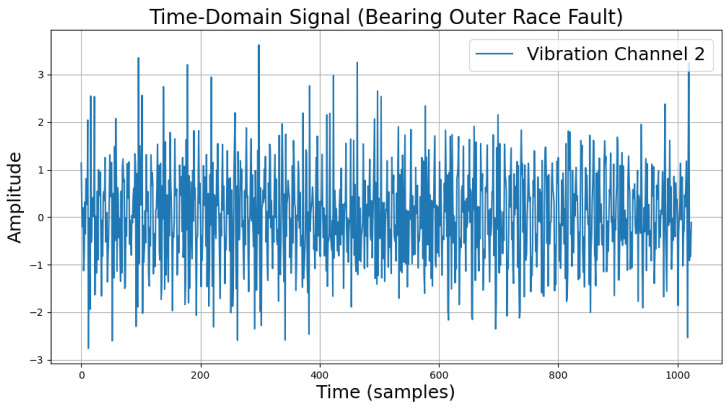 **k**	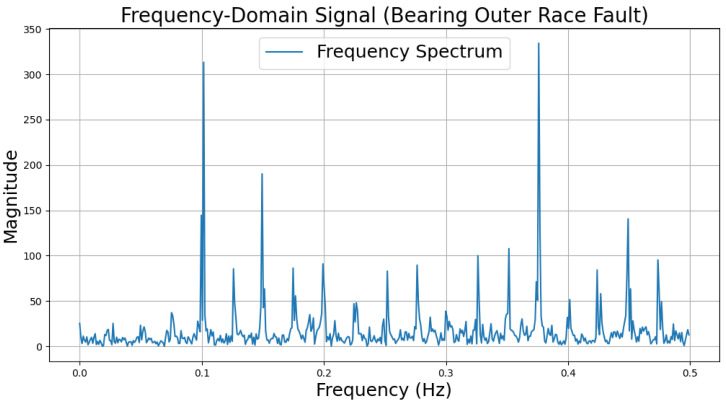 **l**

**Table 3 sensors-25-00009-t003:** Hyperparameter settings.

Hyperparameter	Value
Optimizer	Adam
Loss function	Cross-entropy
Learning rate	0.001
Batch size	64
Number of epochs	100
Normalization method	MinMaxScaler
Number of LSTM units	64
Number of convolutional filters	64
Kernel size	3 and 5
Pooling size	4

**Table 4 sensors-25-00009-t004:** Student model hyperparameters.

Hyperparameter	Value
Optimizer	Adam
Loss function	Soft label cross-entropy + hard label Cross-entropy
Learning rate	0.001
Batch size	64
Number of epochs	50
Normalization method	MinMaxScaler
Number of convolutional filters	32
Kernel size	3
Number of LSTM units	32
Attention layer	Self-attention mechanism
Balancing coefficient (λ)	0.5

**Table 5 sensors-25-00009-t005:** Performance comparison between original and quantized models.

Metric	Original Model	Quantized Model
Average inference time (s)	0.0782	0.0165
Standard deviation of inference time (s)	0.1960	0.0109
Throughput (samples/s)	12.79	60.66
Average memory usage (MB)	501.67	502.38
Average CPU utilization (%)	0.17	0.01
Model size (MB)	0.14	0.02

## Data Availability

The data presented in this study are available upon request from the corresponding author. They are restricted to the experimental results.

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
