# Peer review of "A Deployment Method for Motor Fault Diagnosis Application Based on Edge Intelligence"

_sensors, 2024, doi:10.3390/s25010009_

Round 1
Reviewer 1 Report
Comments and Suggestions for Authors
The manuscript proposes an interesting approach to motor fault diagnosis. I highlight the knowledge distillation and model quantization for reducing the computational complexity of the models. I found it very interesting, highly applicable and efficient for implementation in industrial scenarios. The manuscript presents a good experimental setup with the appropriate performance analysis, including accuracy, memory usage, inference time, and other relevant comparisons.
Although the dataset is described, more details are needed regarding how the signales were adquired are needed. Were these signals adquired in a laboratory under ideal/controlled conditions or in real industrial environments?
Additionally, conducting an experiment under noisy conditions could be very interesting. How would the model perform under different noise levels?
Author Response
Dear Editor/Reviewers,
Please find the attachment for my response letter.
Thank you for your time and consideration. Please do not hesitate to reach out if any further clarification is needed.
Best regards.

Reviewer 2 Report
Comments and Suggestions for Authors
This paper presents a high-quality work on quantization method, which can reduce computational burdening and memory constraints, enabling the applicability and adaptability of resource-limited edge devices to real-time monitoring systems in intelligent manufacturing scenarios. Particularly, detailed descriptions of proposed method through algorithm 1,2 and 3 are clear and the readability was good. Overall work (including structure, intro/background, technical details of proposed method, figures, references, English) is excellent, but I would like to suggest some changes before the final acceptance of the paper.
First, Authors may consider adding some related works on experimental data collection using raspberry pi for machine diagnosis, which are listed below.
- https://doi.org/10.1016/j.mfglet.2024.09.165
- https://doi.org/10.1007/s00170-024-14322-z
Second, texts in Figure 5,6,7,8, 11, 12, 14, and 15 are too small, please revise the font size.
Lastly, the data category that has been used in this work needs to be clearly mentioned. Please add what type of data has been used in this work, and if possible, please add some examples of faulty data that can be mapped to each type of fault.
Author Response

(The authors gave the same response as above.)
